# A Comprehensive Deepfake Detector Assessment Platform

## Abstract

The rapid development of deepfake techniques has raised serious concerns about the authenticity and integrity of digital media. To combat the potential misuse of deepfakes, it is crucial to develop reliable and robust deepfake detection algorithms. In this paper, we propose a comprehensive **D**eepfake **D**etector **A**ssessment **P**latform (**DAP**), covering six critical dimensions: benchmark performance, forgery algorithm generalization, image distortion robustness, adversarial attack resilience, forgery localization accuracy, and attribute bias. Our framework aims to provide a standardized and rigorous approach to assess the performance, generalization ability, robustness, security, localization precision, and fairness of deepfake detection algorithms. Extensive experiments are conducted on multiple public and self-built databases, considering various forgery techniques, image distortions, adversarial attacks, and attributes. The proposed framework offers insights into the strengths and limitations of state-of-the-art deepfake detection algorithms and serves as a valuable tool for researchers and practitioners to develop and evaluate novel approaches in this field. All codes, scripts, and data described in this paper are open source and available at `https://github.com/tempuser4567/DAP`.

## 1 Introduction

The abuse of manipulation and generation technologies will cause chaos in many aspects, such as financial markets. For example, a finance worker was tricked into paying out $25 million to fraudsters in a video conference call generated through deepfake technology (Chen & Magramo, 2024). In order to reduce the negative impact brought by the malicious use of deepfake techniques, many deepfake detection techniques(Li & Lyu, 2018; Yan et al., 2023a; Luo et al., 2021; Qian et al., 2020; Woo et al., 2022; Huang et al., 2022; Haiwei et al., 2022; Li et al., 2020a) have been proposed.

Unfortunately, the detection accuracy of these detectors is actually low. There is a significant gap between the efficacy reported in papers and the actual performance of data in real-world scenarios, which makes it difficult to implement them in practice. Many factors might contribute to this phenomenon. Figure 1 outlines the entire pipeline from fake image and video generation to detection. Three main steps contribute to this phenomenon: (1) Fake Result Generation; (2) Image Distortion; and (3) Adversarial Attacks. **Fake Result Generation**: Based on the type of input data, a proper manipulation or generation algorithm is employed to create a fake face. Any type of deepfake algorithm may potentially be used. **Image Distortion**: Typically, when users upload images or videos on the Internet, social media platforms apply a certain degree of compression, and some users may even use built-in tools to adjust details like contrast. Additionally, some users might add noise, blur, or other distortions to the generated results to avoid detection. **Adversarial Attacks**: Some individuals with advanced technical skills might perform more professional anti-forensic processing on the forged results, such as image reconstruction (Neves et al., 2020) and adversarial attack (Li et al., 2021a). Most forgers do not want their forged fake results to be detected by detection algorithms. Therefore, combining forgery with adversarial attacks poses a significant challenge to detection algorithms. Consequently, in practical detection scenarios, besides basic accuracy, the capabilities of Forgery Algorithm Generalization, Image Distortion Robustness, and Adversarial Attack Resilience are equally important.

Figure 1: Pipeline from deepfake generation to detection

Although some evaluations of detection algorithms exist (Yan et al., 2023b; Pei et al., 2024; Deng et al., 2024), they have several shortcomings: (1) They almost exclusively depend on public databases; (2) They lack actual tests with self-generated fake data; (3) They include too few evaluation aspects.

To more comprehensively assess the detection capabilities of algorithms under various complex conditions, we have built a **D**eepfake **D**etector **A**ssessment **P**latform (**DAP**). The main functions of this platform are: (1) Benchmark Performance Evaluation, (2) Forgery Algorithm Generalization Assessment, (3) Image Distortion Robustness Assessment, (4) Adversarial Attack Resilience Evaluation, (5) Attribute Bias Assessment, (6) Forgery Localization Accuracy Evaluation. Image Distortion Robustness Assessment evaluates the robustness of detection algorithms to common image distortions. The platform applies nine types of image distortions, including adding noise, blur, and compression. Each type consists of 2-8 evaluation tasks. Adversarial Attack Resilience Evaluation assesses the detector's resistance to adversarial attacks, such as image reconstruction. Attribute Bias Assessment examines whether detection algorithms exhibit detection bias toward certain attributes, such as gender. Forgery Localization Accuracy Evaluation assesses the detection algorithm's ability to locate forgeries. It evaluates the accuracy of detecting forged regions in images and forged segments in videos.

This platform serves as a benchmark to continuously validate whether various improvements in detection algorithms are effective under complex conditions. It helps narrow the gap between the detection performance reported in papers and practice. All codes, scripts, and data described in this paper are publicly available at `https://github.com/tempuser4567/DAP`.

The main contributions of this paper are as follows:

- We comprehensively analyzes the entire process from fake image generation to detection, identifying key factors that influence detection, including the diversity of deepfake manipulation algorithms, image distortions applied to forgery results, and adversarial attacks on forgery results.

- The paper designs a platform architecture for evaluating deepfake detection algorithms, comprising six critical dimensions and 27 detail evaluation items. Notably, it includes the category of Large Multimodal Models, expanding the evaluation coverage.

- This paper implements algorithms and scripts for deepfake manipulation, image distortion, and adversarial attack. The platform generates the necessary data (5,976,145 self-generated fake images) for each evaluation category using a unified pipeline, ensuring fair assessment among detection algorithms.

- The paper deploys and evaluates 12 popular detection algorithms, conducting practical evaluations and analyses.

## 2 RELATED WORK

In this section, we introduce deepfake generation techniques, deepfake detection algorithms, and existing evaluation approaches as well as their limitations.

**Deepfake generation technologies** can be categorized into the following four major types: Face Swapping (Li et al., 2019; Chen et al., 2020; Zhu et al., 2021), Facial Reenactment (Thies et al.,

Table 1: Comparison of our work to existing deepfake detection algorithm benchmarks. "Self-augment" means the benchmark contains the augmentation data by itself. "Self-Generate" means the benchmark generates fake data by itself.

| Dimension | | Work (Pei et al., 2024) | DeepfakeBench (Yan et al., 2023b) | Work (Deng et al., 2024) | **Our Paper** |
|---|---|---|---|---|---|
| **Feature** | Dataset | Public Dataset | Public Dataset Self-Augment | Public Dataset Self-Generate Self-Augment | Public Dataset Self-Generate Self-Augment |
| | Evaluation Metrics | 2 | 6 | 1 | 8 |
| | Result Source | Paper | Actual Test | Actual Test | Actual Test |
| | Preprocessing Pipeline | ✗ | ✓ | ✓ | ✓ |
| **Perturbation & Generation** | Self-built Perturbation/ Augmentation | ✗ | 8 | 5 | 9 |
| | Categories of Self-generated Fake Image | ✗ | ✗ | Swapping Reenactment | Swapping Reenactment Attribute Synthesis Large Model |
| | Self-generated Fake Image (Number of Algorithms) | ✗ | ✗ | 25,697 (2) | 5,976,145 (16) |
| **Actual Test** | Generation | ✗ | ✓ | ✓ | ✓ |
| | Image Distortion Robustness | ✗ | ✗ | ✓ | ✓ |
| | Adversarial Attack Resilience | ✗ | ✗ | ✗ | ✓ |
| | Attribute Bias | ✗ | ✗ | ✗ | ✓ |
| | Localization | ✗ | ✗ | ✗ | ✓ |

2016; Hsu et al., 2022; Bounareli et al., 2023), Facial Attribute Manipulation (Liu et al., 2019; Choi et al., 2020; Zhang et al., 2022), and Entire Face Synthesis (Gao et al., 2019; Karras et al., 2019; Skorokhodov et al., 2022). Besides, Large Multimodal Models, which are based on diffusion models, have become popular in recent years for generating images and videos. These include two generation categories: text-to-image and image-to-video. Due to the realistic outputs, they have been widely implemented. Stable Diffusion (Rombach et al., 2022a), Mini-dalle3 (Zeqiang et al., 2023), and Stable Video Diffusion (Blattmann et al., 2023) are widespread algorithms. In addition, **Deepfake Detection algorithms** can be categorized into three types based on detection cues: data-driven detectors (Afchar et al., 2018; Tan & Le, 2019; Nguyen et al., 2019b), spatial artifact-based detectors (He et al., 2019; McCloskey & Albright, 2019; Wang & Chow, 2023), and frequency artifact-based detectors (Qian et al., 2020; Li et al., 2021b; Tan et al., 2024).

Currently, several works have been proposed to survey and evaluate the performance of detection algorithms (Masood et al., 2023; Juefei-Xu et al., 2022; Pei et al., 2024; Seow et al., 2022; Deng et al., 2024; Yan et al., 2023b). Many surveys (Masood et al., 2023; Juefei-Xu et al., 2022; Pei et al., 2024; Seow et al., 2022) categorize deepfake generation and detection algorithms according to their characteristics and summarize the highlights and limitations of each algorithm. Work (Deng et al., 2024) proposes a fair benchmark to measure the performance of a range of detectors. Deep-fakeBench (Yan et al., 2023b) also proposes a unified pipeline for processing public datasets to ensure fairness in evaluating detectors. However, none of these works focus on attribute bias assessment, adversarial attack resilience evaluation, or forgery localization accuracy evaluation. The only study that constructs a private dataset uses just two types of face-swapping on two databases for forgery, without generating other types of fake data as hard examples, and without including fake data from large multimodal models for detector evaluation. Therefore, an up-to-date detection evaluation platform that fully considers current conditions and developments is essential.

## 3 PROPOSED EVALUATION FRAMEWORK

To comprehensively evaluate deepfake detection algorithms, we propose a comprehensive **D**eepfake **D**etector **A**ssessment **P**latform (**DAP**), which covers 27 evaluation tasks related to six critical dimensions. The six dimensions assess the basic performance, generalizability, robustness, security, localizability, and fairness of deepfake detection algorithms.

### 3.1 BENCHMARK PERFORMANCE EVALUATION

In Benchmark Performance Evaluation, we evaluate a detection algorithm through public databases related to four categories.

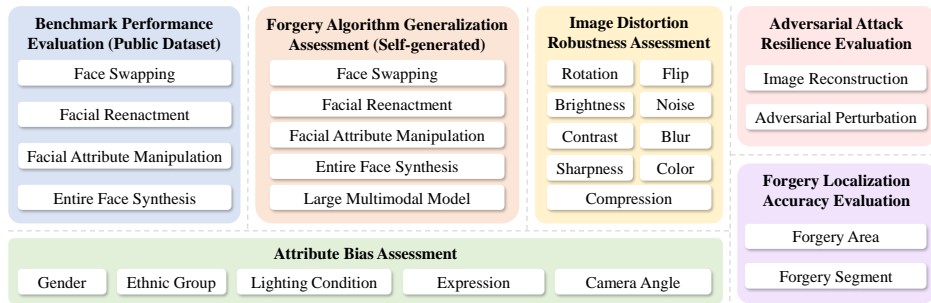

Figure 2: Overview of **D**eepfake **D**etector **A**ssessment **P**latform (**DAP**), which covers 27 evaluation tasks related to six critical dimensions.

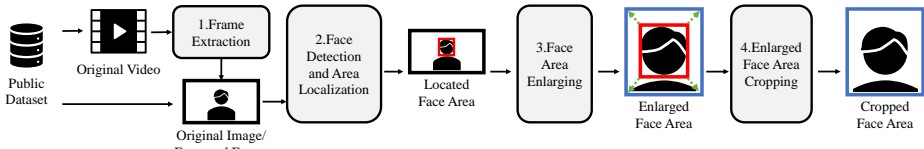

Figure 3: Public dataset pre-processing pipeline

To evaluate the detection algorithm, we first downloaded seven popular public databases from the Internet, including FaceForensics++ (Rossler et al., 2019), Celeb-DF-v1 (Li et al., 2020b), Celeb-DF-v2 (Li et al., 2020b), FakeAVCeleb(Khalid et al., 2021), DeeperForensics-1.0 (Jiang et al., 2020), DFFD (Dang et al., 2020), and DFDC (Dolhansky et al., 2020).

For the sake of fairness, we design a standard data pre-processing pipeline to keep the same evaluation data for each detection algorithm. Figure 3 shows the public database pre-processing pipeline, including 4 steps: (1) Frame Extraction; (2) Face Detection and Face Area Localization; (3) Face Area Enlarging; and (4) Enlarged Face Area Cropping.

By analyzing the detection results, the platform calculates various standardized evaluation metrics for the detection algorithm under different deepfake types, establishing a baseline for detection performance.

### 3.2 FORGERY ALGORITHM GENERALIZATION ASSESSMENT

To evaluate the generalization performance of a deepfake detection algorithm, the evaluation platform proposes Forgery Algorithm Generalization Assessment. Except for four deepfake types in Benchmark Performance Evaluation, Large Multimodal Model is included as the fifth deepfake type. The fake data is entirely generated by the evaluation platform itself.

To achieve this, we select 16 popular deepfake generation and manipulation algorithms, covering the aforementioned five deepfake types. Among them, FaceShifter (Li et al., 2019), FaceDancer (Rosberg et al., 2023), and MobileFaceSwap (Xu et al., 2022) belong to the Face Swapping , while DGFR (Hsu et al., 2022) and HyperReenact (Bounareli et al., 2023) belong to the Facial Reenactment. Additionally, STGAN (Liu et al., 2019) and StarGAN-V2 (Choi et al., 2020) belong to the Facial Attribute Manipulation. StyleGAN2 (Karras et al., 2020), StyleGAN3 (Karras et al., 2021), ProGAN (Karras et al., 2017), and StyleGAN-V (Skorokhodov et al., 2022) belong to the Entire Face Synthesis. LDM (Rombach et al., 2022b), Stable Diffusion (Rombach et al., 2022a), DALLE-mini(Dayma et al., 2021), and DALLE3-mini (Zeqiang et al., 2023) belong to Text-to-Image. Stable-Video-Diffusion (Blattmann et al., 2023) belongs to Image-to-Video.

Figure 4 shows the pipeline of fake data generation and manipulation. To avoid redundant descriptions, we have reclassified the various forgery methods based on the type of input sources into the following six categories: (1) Double Images; (2) Single Image and Single Video; (3) Single Image and Attribute Information; (4) Only Single Image; (5) Only Prompt; and (6) Only Random Number.

The platform generated a total of 5,976,145 fake images. Through the above generation pipeline, the platform simulates the complex forgery situation in the real world. Therefore, this evaluation can test

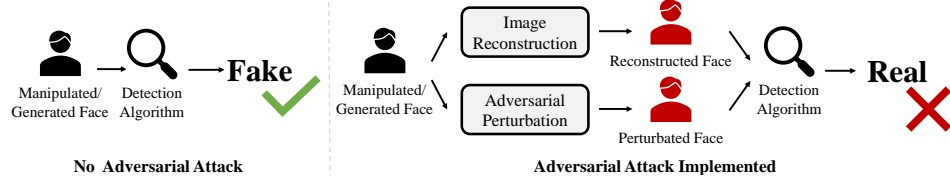

Figure 4: Fake Data Generation Pipeline. Various forgery methods are reclassified into the following six categories based on the type of input.

Figure 5: Pipeline of adversarial attack

the detection algorithm's performance on forgery techniques that may have never been encountered before and obtain more objective generalization evaluation results.

### 3.3 IMAGE DISTORTION ROBUSTNESS ASSESSMENT

Images and videos may undergo certain changes before release, such as adjusting brightness, contrast, color, etc. Besides, some people intentionally add some noise or perform blur processing to evade detection algorithms Therefore, the robustness of a deepfake detection algorithm is very important.

We selected the following nine types as common image disruptions: Compression, Brightness, Contrast, Flip, Rotation, Color, Sharpness, Blur, and Noise. Figure 11 shows the overview of common image distortion for a fake image.

**Compression**: To evaluate the robustness of the detection algorithm to compression, the platform compressed the original images to JPG format in eight degrees: 0.99, 0.95, 0.90, 0.85, 0.80, 0.70, 0.60, and 0.50. **Brightness, Contrast, sharpness, and Color**: These distortions belong to the basic photo color adjustment. For each type of distortion, we take two degrees with greater change effects (-50% and +50%) to make it easier to see whether the detection algorithm can resist the impact of these distortions. **Flip and Rotation**: Flip adopts two methods: from left to right and from top to bottom. The rotation uses four clockwise rotation degrees: 45°, 135°, 225°, and 315°. **Blur and Noise**: These distortions are often used to hide forgery defects and evade deepfake detection. The evaluation platform uses Gaussian, Mean, and Medium filters to achieve blur effects. It also adds noise in the form of Gaussian, Salt and Pepper, and even tokens.

Through the image distortion robustness assessment, the platform can systematically evaluate the detection performance of the algorithm under different types and degrees of distortions, facilitating the analysis of its robustness.

### 3.4 ADVERSARIAL ATTACK RESILIENCE EVALUATION

This section is primarily used to evaluate whether the deepfake detection algorithm can resist adversarial attacks designed to evade deepfake detection.

Figure 5 shows the pipeline of adversarial attack. **Image Reconstruction**: The detection algorithm identifies fake images primarily by detecting forgery traces within the image. Autoencoder can be used to reconstruct an image with minimal visual differences between the original and reconstructed images (Neves et al., 2020). **Adversarial Perturbation**: In contrast to image reconstruction, this

method does not remove the forgery traces but instead interferes with the detection algorithm by adding an adversarial perturbation (Li et al., 2021a), leading to misclassification.

We can compare the evaluation results with those without adversarial attacks and study the defense strategies, such as adversarial training, input conversion, model integration, etc., to improve its robustness against adversarial attacks.

### 3.5 ATTRIBUTE BIAS ASSESSMENT

This section is mainly used to evaluate whether the detection algorithm has a bias toward a certain attribute.

We selected five attributes for evaluation, including Camera Angle, Gender, Ethnic Group, Expression, and Lighting Condition. Each Attribute contains 2-8 different categories. **Camera angle**: Front, Up, Down, Left, Left front, Right front, and Right. **Gender**: Female and Male. **Ethnic group**: Asian East, Asian South, African, Caucasian Europe, and Caucasian American. **Expression**: Neutral, Distust, Sad, Surprise, Contempt, Angry, Fear, and Happy. **Lighting Condition**: Left up, Up, Right up, Left, Uniform, Right, Left down, Down, Right down

This evaluation helps identify biases and weaknesses of the detection algorithm concerning specific attributes. Consequently, strategies to mitigate these biases, such as data balancing and attribute-aware training, can be proposed and validated.

### 3.6 FORGERY LOCALIZATION ACCURACY EVALUATION

This section mainly evaluates the forgery localization ability of detection algorithms.

The platform uses data from the FaceForensics++ database to evaluate the manipulation region localization and from the Lav-DF database to evaluate video manipulation segment localization. This evaluation uses specialized metrics to quantify the algorithm's manipulation localization performance, including IoU (Intersection over Union), pixel accuracy, AP@IoU threshold (Average Precision at a specific IoU threshold), and AR@IoU threshold (Average Recall at a specific IoU threshold).

Through this evaluation, the platform can analyze the localization ability of detection algorithms under different manipulation techniques, forgery regions and shapes.

## 4 EXPERIMENTS AND RESULTS

In this section, we introduce the experimental setup, evaluation results of each evaluation category, and insights behind the results.

### 4.1 EXPERIMENTAL SETUP

The experimental setup includes four parts: (1) Detection Algorithms; (2) Databases and Pre-processing; (3) Evaluation Data Sampling; and (4) Evaluation Metrics. All experiments were conducted on a server running Ubuntu 20.04, equipped with an NVIDIA A100 GPU (40GB memory).

**Detection Algorithms**: In this experiment, we evaluated 12 deepfake detection algorithms, including Xception (Rossler et al., 2019), SRM (Luo et al., 2021), SBI (Shiohara & Yamasaki, 2022), DSP-FWA (Li & Lyu, 2018), Multiple-Attention (Zhao et al., 2021), Efficientnetb4 (Tan & Le, 2019), MesoInception (Afchar et al., 2018), MesoNet (Afchar et al., 2018), UCF (Yan et al., 2023a), CNN-Aug (Wang et al., 2020), ClassNSeg (Nguyen et al., 2019a), and BA-TFD+ (Cai et al., 2023).

**Databases and Pre-processing**: The platform implements 11 popular public databases, including FaceForensics++ (Rossler et al., 2019), Celeb-DF-v1 (Li et al., 2020b), Celeb-DF-v2 (Li et al., 2020b), FakeAVCeleb(Khalid et al., 2021), DeeperForensics-1.0 (Jiang et al., 2020), DFFD (Dang et al., 2020), DFDC (Dolhansky et al., 2020), CelebA (Liu et al., 2015), VGGface2 (Cao et al., 2018), VidTIMIT (Sanderson & Lovell, 2009), and Lav-DF (Cai et al., 2023). These databases cover data with labels of all four deepfake categories. In addition, it contains special data with labels of the forgery region, the forgery segment, and different attributes. The platform followed the

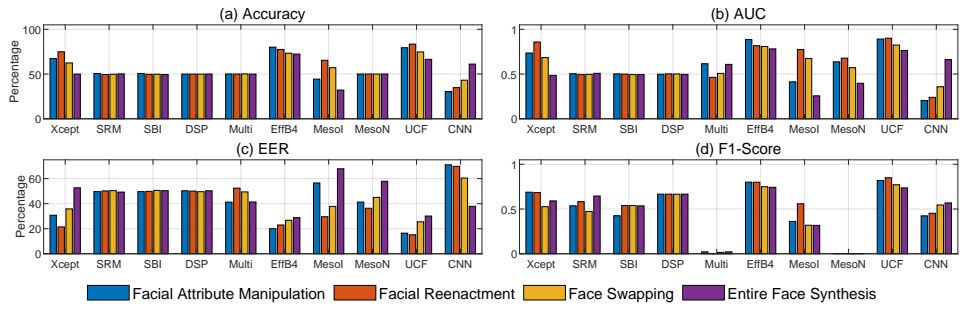

Figure 6: Benchmark Performance Evaluation: comparison of Accuracy, AUC, EER, and F1-Score among 10 deepfake detection algorithms.

pipeline described in Section B.2.2, and generated over 5 million fake images. The platform applied common image distortions to the data from the public database. Each real and fake image undergoes nine types of distortions. The platform also used adversarial attack algorithms to reconstruct and add adversarial perturbations to fake images.

**Evaluation Data Sampling**: The platform includes six evaluation categories, 27 evaluation subcategories, and over 40 sub-tasks. To ensure evaluation efficiency, we set 10,000 test samples for each sub-task, with 5,000 being real samples and 5,000 being fake samples. Additionally, to avoid test data bias caused by differences in the number of samples across databases, we sampled an equal number of data from each database to form the evaluation set.

**Evaluation Metrics**: We adopt a variety of standardized evaluation metrics to comprehensively measure the performance of each detection algorithm, including Accuracy (Acc), Precision (Pre), Recall (Rec), F1-score (F1), Area Under the ROC Curve (AUC), Equal Error Rate (EER), ROC curve, Precision-Recall Cure, and confidence difference.

## 4.2 BENCHMARK PERFORMANCE EVALUATION

This experiment is to evaluate the benchmark performance of detection algorithms on public datasets.

To simulate the most realistic evaluation scenarios, we use the pre-trained models for each detector. Figure 6 shows the performance of Accuracy, AUC, F1-score, and EER among 10 detectors. Overall, the accuracy of the detection algorithms is generally low. The highest accuracy achieved was 83.43%, while the lowest was only 30.4%. Half of the detection algorithms have an accuracy around 50%, regardless of the deepfake type of evaluation. The other half of the detectors show significant accuracy variations depending on the deepfake type.

These experimental results reveal the basic detection performance of the algorithms. For other evaluation metrics results, please refer to Appendix G.0.1.

## 4.3 FORGERY ALGORITHM GENERALIZATION ASSESSMENT RESULTS

This assessment evaluates the generalization ability of detectors to unknown deepfake algorithms and data domains.

The platform deployed 16 deepfake algorithms, covering various deepfake categories. Figure 7 shows the histogram results of the evaluation. Overall, the accuracy of the detectors remains generally low, and most detectors achieve an accuracy around 50%. The highest and lowest accuracies are 80.42% and 36.52%, respectively. Only EfficientNetB4 and UCF performed relatively well compared to the other eight detectors. Additionally, the Large Multimodal Model is the second most effectively detected category by them.

These experimental results highlight the detection generalization capabilities of each detector for different forgery categories. For other evaluation metrics results, please refer to Appendix G.1.

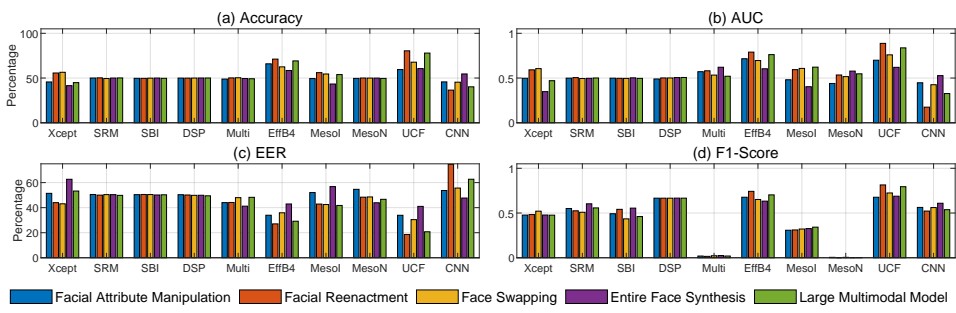

Figure 7: Forgery Algorithm Generalization Assessment: comparison of Accuracy, AUC, EER, and F1-Score among 10 deepfake detection algorithms.

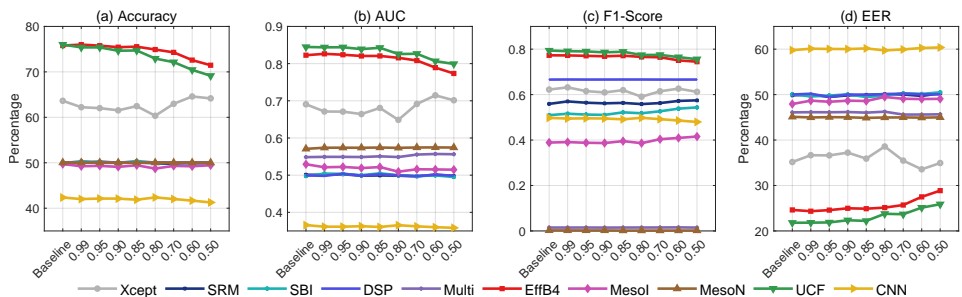

Figure 8: Image Distortion Robustness Evaluation (Compression): comparison of Accuracy, AUC, EER, and F1-Score among 10 deepfake detection algorithms.

### 4.4 IMAGE DISTORTION ROBUSTNESS ASSESSMENT RESULTS

This assessment includes nine types of distortions and is used to evaluate whether the detectors are susceptible to common image distortions.

We present the results of Compression. For better comparison, the results for the four deepfake categories are averaged in this section. Figure 8 shows the trend of evaluation metrics as the compression ratio decreases. Overall, detectors with initially higher accuracy show a gradual decrease in accuracy as the compression ratio decreases. EfficientNetB4 and UCF remain the strongest detectors, consistently achieving the top two positions across the four metrics under various compression ratios. For results of other distortions, please refer to Appendix G.2.

### 4.5 ADVERSARIAL ATTACK RESILIENCE EVALUATION

This part evaluates whether the detectors are capable of resisting adversarial attacks. For Images Reconstruction, the platform uses the GANprintR algorithm to remove GAN "fingerprints" from synthetic fake images. Figure 9 shows that Xception and UCF suffers GANprintR but EfficientNetB4 does not. For Adversarial Perturbation, the platform employs the StyleAttack algorithm to generate anti-forensic fake face images.

This evaluation verifies whether detectors have the capability to resist adversarial attacks. For adversarial perturbation and other evaluation metrics results, please refer to Appendix G.3.

### 4.6 FORGERY LOCALIZATION ACCURACY EVALUATION

This evaluation aims to fulfill specific evaluation requirements for forgery localization, focusing on forgery regions and forgery segments. For forgery region evaluation, Table 2 presents the evaluation results of the ClassNSeg algorithm. While the recall reaches 86.95%, the accuracy and precision are relatively low. For forgery segment evaluation, Table 3 shows that BA-TFD+ algorithm achieves a desirable average precision (AP) and average recall (AR).

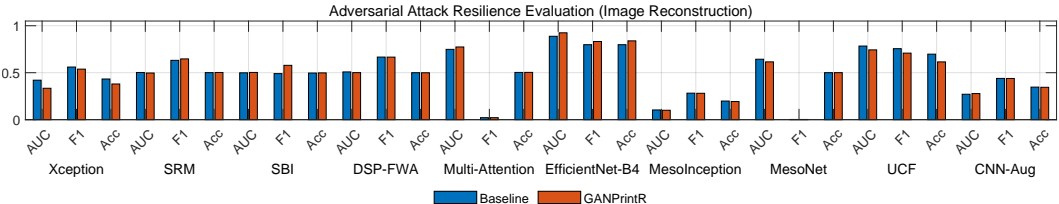

Figure 9: Adversarial Attack Resilience Evaluation (Image Reconstruction): comparison of AUC, F1-Score, and Acc among 10 deepfake detection algorithms.

Table 2: Evaluation Results of Forgery Region Location.

| Acc | AUC | EER | Precision | Recall | F1-Score | Conf-Diff |
|-----|-----|-----|-----------|--------|----------|-----------|
| 51.28% | 0.5279 | 48.32% | 50.75% | 86.94% | 0.6409 | 0.7674 |

Table 3: Evaluation Results of Video Segment Location.

| Average Precision | | | Average Recall | | | |
|-------------------|--------|--------|--------|--------|--------|--------|
| AP@0.5 | AP@0.75 | AP@0.95 | AR@100 | AR@50 | AR@20 | AR@10 |
| 97.68% | 83.53% | 3.38% | 80.59% | 79.15% | 77.78% | 77.35% |

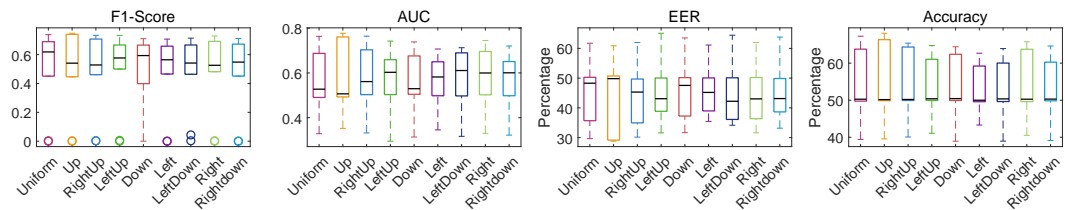

Figure 10: Attribute Bias Evaluation (Lighting Condition): comparison of F1-Score, AUC, EER and Accuracy among 10 deepfake detection algorithms.

### 4.7 SCENE ATTRIBUTE BIAS ASSESSMENT

This assessment focuses on assessing the existence of attribute bias in the detectors. The platform evaluated five attributes and we present the experimental results for the Lighting Condition attribute. Figure 10 presents the evaluation results for Lighting Condition. It is evident from the figure that the black median lines corresponding to different lighting conditions vary significantly, indicating a clear bias. The box range reveals that the UP category has the highest F1-score, AUC, accuracy, and the lowest EER. This evaluation can determine whether a detector exhibits bias towards a specific attribute. For results of other attributes, please refer to Appendix G.4.

## 5 CONCLUSION

This paper analyzes two issues: (1) the difficulty in reproducing experimental results and (2) the low detection accuracy in real-world scenarios. Through a comprehensive review of the entire process from image forgery to detection, we have identified potential causes of these issues and accordingly established a comprehensive and fair deepfake detector evaluation platform. This platform evaluates detectors from six major dimensions, deploying 16 deepfake algorithms involving five main categories to generate a large amount of fake data for generalization assessment. Especially, five large multimodal models are included. Additionally, the platform is equipped with adversarial attack-related algorithms to practically test the detectors' resistance. Finally, considering the practical usage of detectors, the platform introduces attribute bias and forgery localization assessments, both of which have been tested and analyzed. As deepfake algorithms continue to proliferate and evasion techniques mature, the demand for forgery localization of fake videos and images will become more urgent. Therefore, the platform can serve as a benchmark to drive the development of deepfake detectors toward more practical applications.

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

APPENDIX

# A    RELATED WORK

In this section, we introduce deepfake generation techniques, deepfake detection algorithms and existing evaluation approaches and their limitations.

## A.1    DEEPFAKE GENERATION TECHNIQUES

Deepfake generation technologies can be categorized into the following four major types: Face Swapping, Facial Reenactment, Facial Attribute Manipulation, and Entire Face Synthesis. **Face Swapping** is used to replace face A in the target image with face B in the source image. The most popular face-swapping algorithms are FaceShifter (Li et al., 2019), FSGAN (Nirkin et al., 2019), SimSwap (Chen et al., 2020), and MegaFS (Zhu et al., 2021). **Facial Reenactment** is used to reenact the facial expressions and movements of a target video on the face in a source image. Face2Face (Thies et al., 2016), Neural Textures (Thies et al., 2019), DG (Hsu et al., 2022), and HyperReenact (Bounareli et al., 2023) are classical algorithms for achieving Facial Reenactment. **Facial Attribute Manipulation** is also known as face editing. It is used to modify the attributes of a human face, including gender, hair color, age, etc. Many studies pay attention to this category, such as STGAN (Liu et al., 2019), StarGAN2 (Choi et al., 2020), and FDNeRF (Zhang et al., 2022). **Entire Face Synthesis** is used to generate non-existent human face images or videos. The classical algorithms are ProGAN (Gao et al., 2019), StyleGAN (Karras et al., 2019), StyleGAN-v (Skorokhodov et al., 2022). Please note that Entire Face Synthesis can generate videos, such as StyleGAN-v (Skorokhodov et al., 2022). **Large Multimodal Models**, which are based on diffusion models, have become popular in recent years for generating images and videos. These include two generation categories: text-to-image and image-to-video. Due to their realistic effects, they are widely used in face synthesis. Stable Diffusion (Rombach et al., 2022a), Mini-dalle3 (Zeqiang et al., 2023), and Stable Video Diffusion (Blattmann et al., 2023) are widespread Algorithms.

## A.2    DEEPFAKE DETECTION ALGORITHMS

Detection algorithms can be categorized into three types based on detection cues: data-driven detectors, spatial artifact-based detectors, and frequency artifact-based detectors. **Data-driven detectors** use a large dataset of real and fake data (images and videos) to train the detection algorithm to learn the differences between real and fake images. The well-trained model is then used to detect the authenticity of images. Classical detectors are MesoNet (Afchar et al., 2018), EfficientNet-B4 (Tan & Le, 2019), Capsule-forensics (Nguyen et al., 2019b), and Xception (Rossler et al., 2019). This type of detector is relatively simple in design but heavily relies on the training data. If the test data significantly differs from the training data, the detection performance will degrade substantially. **Spatial artifact detectors** take image inconsistencies as clues for detection. Techniques related to Face Swapping, Facial Reenactment, and Facial Attribute Manipulation usually only forge the facial region of an image, leaving other non-facial background areas unchanged. This leads to inconsistencies between the modified and unmodified regions. Many inconsistencies have already been studied, such as color space (He et al., 2019), saturation cues (McCloskey & Albright, 2019), and noise (Wang & Chow, 2023). The advantage of these detectors lies in their stronger generalization capabilities compared to data-driven detectors. **Frequency artifact detectors** first transform images from the time domain to the frequency domain. Then, they draw the detection conclusion by examining whether the frequency domain features are abnormal. There are many studies in this field, such as F3-Net (Qian et al., 2020), FDFL (Li et al., 2021b), and FreqNet (Tan et al., 2024).

## A.3    EXISTING EVALUATION APPROACHES AND THEIR LIMITATIONS

Currently, several studies have been proposed to survey and evaluate the performance of detection algorithms (Masood et al., 2023; Juefei-Xu et al., 2022; Pei et al., 2024; Seow et al., 2022; Deng et al., 2024; Yan et al., 2023b)

Studies (Masood et al., 2023; Juefei-Xu et al., 2022; Pei et al., 2024; Seow et al., 2022) are surveys on deepfake. These studies organize deepfake generation and detection algorithms, categorize them

according to their characteristics, and summarize the highlights and limitations of each algorithm. However, the authors compare and analyze detection algorithms only from the performance results recorded in the corresponding papers.

Research (Deng et al., 2024) finds that evaluation performance varies for the same detector and database. This inconsistency may be due to the varying conditions between studies. Therefore, they propose a fair benchmark to measure the performance of a range of detectors. The authors also generate self-generated examples using two face-swapping algorithms to build a private dataset containing 25,697 fake images as hard examples for detector evaluation.

DeepfakeBench (Yan et al., 2023b) also proposes a unified pipeline for processing public datasets to ensure fairness in evaluating detectors. In addition, the authors advocate for standardized evaluation metrics and protocols to enhance transparency and reproducibility. In the Evaluation section, they conduct practical tests on domain and manipulation generalization using processed data and assess robustness against image augmentation.

However, none of these studies focus on attribute bias assessment, adversarial attack resilience evaluation, or forgery localization accuracy evaluation. The only study that constructs a private dataset uses just two types of face-swapping on two databases for forgery, without generating other types of fake data as hard examples, and without including fake data from large multimodal models for detector evaluation. Therefore, an up-to-date detection evaluation platform that fully considers current conditions and developments is essential.

## B    PROPOSED EVALUATION FRAMEWORK

To comprehensively evaluate deepfake detection algorithms, we propose a comprehensive **D**eepfake **D**etector **A**ssessment **P**latform (**DAP**), which covers 27 evaluation tasks related to six critical dimensions. The six dimensions are Benchmark Performance Evaluation, Forgery Algorithm Generalization Assessment, Image Distortion Robustness Assessment, Adversarial Attack Resilience Evaluation, Forgery Localization Accuracy Evaluation, and Attribute Bias Assessment. The six dimensions assess the basic performance, generalizability, robustness, security, localizability, and fairness of deepfake detection algorithms. Each dimension consists of 2-8 evaluation tasks. The platform prepares task-specific data for each task through the corresponding strategy.

### B.1    BENCHMARK PERFORMANCE EVALUATION

In Benchmark Performance Evaluation, we evaluate a detection algorithm through public databases related to four categories: Face Swapping, Facial Reenactment, Facial Attribute Manipulation, and Entire Face Synthesis. We will introduce the evaluation of benchmark performance from the following two aspects: (1) Public Databases and Pre-processing and (2) Standardized Evaluation Metrics.

### B.1.1    PUBLIC DATABASES AND PREPROCESSING

To evaluate the detection algorithm, we first downloaded seven popular public databases from the Internet, including FaceForensics++ (Rossler et al., 2019), Celeb-DF-v1 (Li et al., 2020b), Celeb-DF-v2 (Li et al., 2020b), FakeAVCeleb(Khalid et al., 2021), DeeperForensics-1.0 (Jiang et al., 2020), DFFD (Dang et al., 2020), and DFDC (Dolhansky et al., 2020).

Public databases are various from each other. The original data may be videos or images. Some of the images are full of faces, but others have large areas of background. For a piece of video, it is common to divide it into a series of frames and save them as images. Then a face area is cropped from a frame if the face area is far smaller than the area of the frame. Finally, the detection algorithm judges whether the face area is generated. Although most people follow the above three steps to perform detection, the experiment results are different seriously for the same detection and public database. The main reason is the details of database pre-processing. When a video is divided into frames, we can choose I-, P-, or B-frame as the target frame and decide the interval for video frame extraction. When a face area is cropped from an image or a frame, many face detection and location algorithms can be selected. The relative cropping area for a single face may not follow a unified standard.

### B.1.2    PROPOSED PUBLIC DATABASE PRE-PROCESSING PIPELINE

For the sake of fairness, it is important to keep the same evaluation data for each detection algorithm. We analyze different public databases and design a standard data pre-processing pipeline that covers videos and images. Through this pipeline, we process seven selected databases and save all results into images of faces.

Figure 3 shows the public database pre-processing pipeline, including 4 steps: (1) Frame Extraction; (2) Face Detection and Face Area Localization; (3) Face Area Enlarging; and (4) Enlarged Face Area Cropping.

The original data selected from a public database may be an original image or an original image. **Step 1: Frame Extraction** For a video, the pipeline performs Frame Extraction, and gets a series of frames. FFmpeg libraries are implemented during this step because FFmpeg is one of the most popular tools for dealing with videos. Please note that the pipeline does not consider the class (I-, P-, or B-frame) of extracted frames. Instead, the pipeline extracts a frame every n frames. As a evaluation platform, it is important to avoid data bias. If the pipeline extracts frames all from one class, such as I-frame. The deepfake detection algorithm may enhance the detection only for I-frames but ignore the performance of P- and B-frames. The evaluation results may seem good. In fact, when facing other manipulated videos with forgeried frames of B- or P-frames, the performance of this detection algorithm will drop. The parameter n can be set for different databases according to the practicality and kept secret.

**Step 2: Face Detection and Face Area Localization** The pipeline detects the face in an original image from public databases or a frame extracted from a video. If a face is detected, the pipeline will get its location. The pipeline locates faces through Retaniface algorithm which is one of the most popular human face detection algorithms.

**Step 3: Face Area Enlarging** After getting the face location, the pipeline enlarges the scope of the detected face area through Faceinsight. Otherwise, the original detected face area could not include the outline of the face, such as ears. The red bounding box is the detected face area and the blue bounding box is the enlarged face area.

**Step 4: Enlarged Face Area Cropping** The pipeline crops the enlarged face area from an image or an extracted frame. Then the pipeline saves the cropped image in a lossless compression format, such as PNG.

Through the proposed public database pre-processing pipeline, both videos and images can be processed and saved in a unified format.

### B.1.3    STANDARDIZED EVALUATION METRICS

We adopt a variety of standardized evaluation metrics to comprehensively measure the performance of each detection algorithm, including Accuracy (Acc), Precision (Pre), Recall (Rec), F1-score (F1), Area Under the ROC Curve (AUC), Equal Error Rate (EER), ROC curve, Precision-Recall Cure, and confidence difference.

### B.1.4    DATA PREPARATION AND EVALUATION

To evaluate the detection performance of the detection algorithm for different types of deepfake, the evaluation platform categorizes various public databases into four types: Face Swapping, Facial Reenactment, Facial Attribute Manipulation, and Entire Face Synthesis. Each type may contain multiple public databases. For example, Face Swapping contains six public databases: FaceForensics++, Celeb-DF-v1, Celeb-DF-v2, FakeAVCeleb, DeeperForensics-1.0, and DFDC. During the evaluation, for each deepfake category, the platform feeds the pre-processed public database data into the detection algorithm and obtains the corresponding evaluation results. By analyzing the detection results, the platform calculates various standardized evaluation metrics for the detection algorithm under different deepfake types, establishing a baseline for detection performance.

## B.2 FORGERY ALGORITHM GENERALIZATION ASSESSMENT

To evaluate the generalization performance of a deepfake detection algorithm, the evaluation platform proposes Forgery Algorithm Generalization Assessment. Five deepfake types are included in this evaluation. Except for four deepfake types in Benchmark Performance Evaluation, Large Multimodal Model is included as the fifth deepfake type. Similar to Benchmark Performance Evaluation, each deepfake type corresponds to specific evaluation data. However, in this case, the fake data is entirely generated by the evaluation platform itself.

### B.2.1 FAKE TYPES AND FORGERY ALGORITHMS

To achieve this, we select 16 popular deepfake generation and manipulation algorithms, covering the aforementioned five deepfake types. These algorithms were successfully deployed, and the corresponding fake data were generated and saved. For **Face Swapping**, we choose FaceShifter, FaceDancer, and MobileFaceSwap as the manipulation algorithm. The evaluation platform randomly selects a pair of human face images and gets a face-swapped image through a Face Swapping algorithm. For **Face Reenactment**, we choose HyperReenact and DGFR as the manipulation algorithm. The evaluation platform randomly selectes a human face image (FA) and a human face video (HV) as a source-target pair. Then the platform reenacts the facial movement of the human face video (FV) for the human face image (FA). For **Facial Attribute Manipulation**, we choose StarGAN-2 and STGAN as the manipulation algorithm. The evaluation platform randomly selects a pair of human face images and transforms the style from image A to image B with StarGAN-2. For STGAN, the evaluation platform randomly selects a human face image and the attribute needs to be manipulated. Then an attribute-manipulated human face image is generated and saved. For **Entire Face Synthesis**, we choose ProGAN, StyleGAN-2, StyleGAN-3, and StyleGAN-V as the generation algorithm. The input of these deepfake generation algorithms is a random number. The output of the first three algorithms is human face image. The output of StyleGAN-V is a video. For **Large Multimodal Model**, we choose Stable Diffusion, DALLE mini, LDM, DALLE3 mini, and Stable Video Diffusion as the generation algorithm. The first four algorithms can generate images according to prompts. For example, the platform can input a prompt "photo of a smiling young woman" into these algorithms. Then a photo of a young female with a smiling expression will be obtained. Stable Video Diffusion can generate a video using an image.

### B.2.2 FAKE DATA GENERATION PIPELINE

After deploying the deepfake generation and manipulation algorithms, the platform can use images and videos from public databases as sources for various types of forgeries. Figure 4 shows the pipeline of fake data generation and manipulation. To avoid redundant descriptions, we have reclassified the various forgery methods based on the type of input sources into the following six categories: (1) **Double Images**: Two images are inquired to perform a manipulation or generation, such as part of algorithms of Face Swapping and Facial Attribute Manipulation. (2) **Single Image and Single Video**: When performing a Manipulation or fake data generation, a human face image and a video are needed as the input data. It is usually used for algorithms of Facial Reenactment. (3) **Single Image and Attribute Information**: The input of manipulation requires not only an image but also attribute information. The algorithm modifies the attribute of the human face in the image according to the attribute information. For example, the platform can change the hair color from brown to black and keep other attributes remain. (4) **Only Single Image**: No other inputs are required except for a human face image. It is common for algorithms of image-to-video, which is categorized into Large Multimodal Model in Section B.1.3. (5) **Only Prompt**: No image or video is needed as the input for fake data generation. Algorithms generate fake data only through the input prompts. (6) **Only Random Number**: Similar to Only Prompt, there is no need to prepare images or videos for fake data generation. A random number is the only input for image or video generation, which is common for Entire Face Synthesis.

All image outputs are saved in lossless compression format. For generated videos, the platform extracts frames through FFmpeg.

### B.2.3 EVALUATION DATA PREPARATION

For generalization Evaluation, the platform selects 10 public databases: FaceForensics++, Celeb-DF-v1, Celeb-DF-v2, FakeAVCeleb, DeeperForensics-1.0, DFFD, DFDC, VGGFace2, CelebA, and VidTIMIT. For each deepfake type, the Evaluation platform prepares real data original from public databases and fake data generated itself. Please note that the original fake data of all public databases are ignored in this evaluation.

**Real data**: For source image-specific manipulation (Double Images, Single Image and Single Video, Single Image and Attribute Information, Only Single Image), the real data can be selected from the database of the source image. Because the source images and the manipulated images are original from the same data domain, the difference between the two groups of images is slight. As a result, this is the most challenging setting.

For no source image-specific fake data generation (Only Prompt and Only Random Number), there is no corresponding real data. The evaluation platform uniformly samples real data from 10 public databases as representative real data in the wild.

**Fake data**: For source image-specific manipulation, the platform uses different deepfake algorithms to generate fake data based on real data from public databases. Facial reenactment needs both images and videos. Part of databases can not be manipulated, because they only contain images without videos.

For no source image-specific fake data generation, the platform generates a large amount of fake data through different random numbers and prompts.

### B.2.4 GENERALIZATION ABILITY EVALUATION

Through the above generation pipeline, the platform obtains the prepared real and fake data for each deepfake type. These data are sourced from different databases and cross-manipulated through different algorithms. To a large extent, it simulates the complex forgery situation in the real world. Therefore, this evaluation can test the detection algorithm's performance on forgery techniques that may have never been encountered before and obtain more objective generalization evaluation results. The evaluation metrics are those mentioned in Section B.1.3.

## B.3 IMAGE DISTORTION ROBUSTNESS ASSESSMENT

The robustness of a deepfake detection algorithm is very important, because the data to be detected may undergo unexpected distortions which affects the detection results. The images and videos spread on the internet, whether real or fake data, are likely to undergo a certain degree of compression. The compressed data is different from the original data. Images and videos may undergo certain optimizations before release, such as adjusting brightness, contrast, color, etc. Besides, some people intentionally add some noise or perform blur processing to evade detection algorithms

### B.3.1 COMMON IMAGE DISTORTIONS

We have analyzed and listed various possible image distortions in reality, and selected the following 9 types as common image disruptions: Compression, Brightness, Contrast, Flip, Rotation, Color, Sharpness, Blur, and Noise. Figure 11 shows the overview of common image distortion for a fake image. Please note, image distortion affects not only fake images but also real images, such as Compression on the Internet. Therefore, the platform performs image common distortion both for real and fake images.

**Compression**: Compression is an important factor affecting the robustness of detection algorithms. Because the compressed data and the original data are usually not within the same data domain, learning-based detection algorithms may not have learned the difference between compressed real and fake data. As a result, the detection algorithm will perform high false or missed detection rates. To evaluate the robustness of the detection algorithm to compression, the platform compressed the original images to JPG format in eight degrees: 0.99, 0.95, 0.90, 0.85, 0.80, 0.70, 0.60, and 0.50. We can observe how the algorithm's robustness changes with the compression rate according to the series of results.

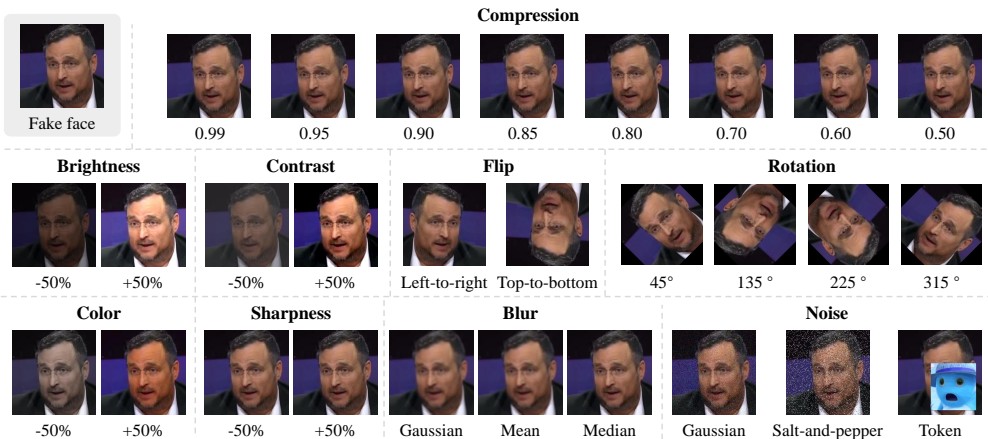

Figure 11: Overview of Common Image Distortions

**Brightness, Contrast, sharpness, and Color**: These distortions belong to the basic photo color adjustment and are often used to beautify photos and enhance their appearance. Manipulated and generated images often require further modification to make it difficult for the human eye to detect. There are many degrees of these four types of image distortions. For each type of distortion, we take two degrees with greater change effects (-50% and +50%) to make it easier to see whether the detection algorithm can resist the impact of these distortions.

**Flip and Rotation**: Most human faces in public databases are in a front direction, with little rotation or flipping. However, in practice, many facial regions in the tested image are not in the front direction. If the detection algorithm does not consider situations other than the front direction during training, it may not be able to accurately perform detection. Flip adopts two methods: from left to right and from top to bottom. The rotation uses four clockwise rotation degrees: 45°, 135°, 225°, and 315°.

**Blur and Noise**: These distortions are not commonly used in image beautification, but are often used to hide forgery defects and evade deepfake detection. Due to the poor adaptability of the manipulation algorithm to the target facial image, some of the fake faces have some easily detectable flaws. Therefore, in order to make the forged result look more like a natural face, the person who created the fake image will blur the forged result and hide the manipulated details. The evaluation platform uses Gaussian, Mean, and Medium filters to achieve blur effects. To prevent the fake face from being detected by the detection algorithms, the person who manipulated the image will add noise to the image to interfere with the deepfake detection algorithm and attempt to escape the forgery detection. To simulate this situation, we added noise in the form of Gaussian, Salt and Pepper, and even tokens.

### B.3.2 EVALUATION DATA PREPARATION

To evaluate the robustness of the detection algorithm against the aforementioned common image distortions, the platform needs to generate nine types of distortion results for the existing real and fake data. In this section, the platform uses the pre-processed public database data from Section B.1.3 as the source data for distortion.

### B.3.3 ROBUSTNESS EVALUATION UNDER DISTORTIONS

The platform inputs data processed with common image distortions into the deepfake detection algorithm under test. Then the evaluation results across 28 sub-items within 9 major categories are obtained. Then, the evaluation metrics for each type of distortion are calculated using the method described in Section B.1.3 Finally, by comparing the evaluation results before and after common image distortion, we can analyze the trends and degrees of change to draw evaluation conclusions.

Through the image distortion robustness assessment, the platform can systematically evaluate the detection performance of the algorithm under different types and degrees of distortions, facilitating the analysis of its robustness. This assessment can also be used to evaluate the effectiveness of strategies against image distortions, such as data augmentation and adversarial training, thereby improving the algorithm's stability in real-world applications.

### B.4 Adversarial Attack Resilience Evaluation

This section is primarily used to evaluate whether the deepfake detection algorithm can resist adversarial attacks designed to evade deepfake detection. We have discussed some methods, such as blur, to hide forgery defects and fool deepfake detection algorithm in Section B.3.1. But those are only based on common image distortions. In this section, the platform addresses more advanced learning-based adversarial attacks, including image reconstruction and adversarial perturbation attacks.

#### B.4.1 Adversarial Attack Generation

Figure 5 shows the pipeline of adversarial attack. When there are no adversarial attacks, a manipulated or generated face image is easy to detect as a fake image. However, when adversarial attacks are implemented, images that should be detected as fake may be incorrectly classified as real.

Both image reconstruction and adversarial perturbation attacks have two main objectives: (1) The image after the attack should appear as similar as possible to the image before the attack. (2) The detection algorithm should be highly likely to classify the fake image as a real image.

**Image Reconstruction**: The detection algorithm identifies fake images primarily by detecting forgery traces within the image. If these traces are removed, the fake images can evade deepfake detection. Autoencoder can be used to reconstruct an image with minimal visual differences between the original and reconstructed images (Neves et al., 2020).

**Adversarial Perturbation**: In contrast to image reconstruction, this method does not remove the forgery traces but instead interferes with the detection algorithm by adding an adversarial perturbation (Li et al., 2021a), leading to misclassification. The adversarial perturbation can be specifically generated for a particular detection algorithm, or it can be a general adversarial perturbation with strong generalization capabilities. Please note that the adversarial perturbation can be added not only after the fake image is generated, but also within the latent space layer.

#### B.4.2 Evaluation Data Preparation

To facilitate comparison with performance metrics before the adversarial attack, the platform uses the pre-processed public database data from Section B.1.3. Unlike the image common distortion in Section B.3.2, the platform does not need to perform adversarial attacks on all data, because the real image is real and does not require additional processing. Therefore, only the forged data is processed. The relevant data from the public databases is subjected to image reconstruction and the addition of adversarial perturbation. These adversarial forged data are then combined with the real data to form the evaluation database.

#### B.4.3 Resilience Evaluation Against Attacks

The platform evaluates the detection algorithm using organized evaluation data and obtains its evaluation results for different types of adversarial attacks. Then, we can compare the evaluation results with those without adversarial attacks and study the defense strategies, such as adversarial training, input conversion, model integration, etc., to improve its robustness against adversarial attacks.

### B.5 Attribute Bias Assessment

This section is mainly used to evaluate whether the detection algorithm has a bias toward a certain attribute. The detection algorithm may not only have a bias in deepfake type of test data but may also have a bias in certain attributes. For example, when a detection algorithm is trained on a database consisting entirely of males, it may not be possible to accurately perform a detection when the test

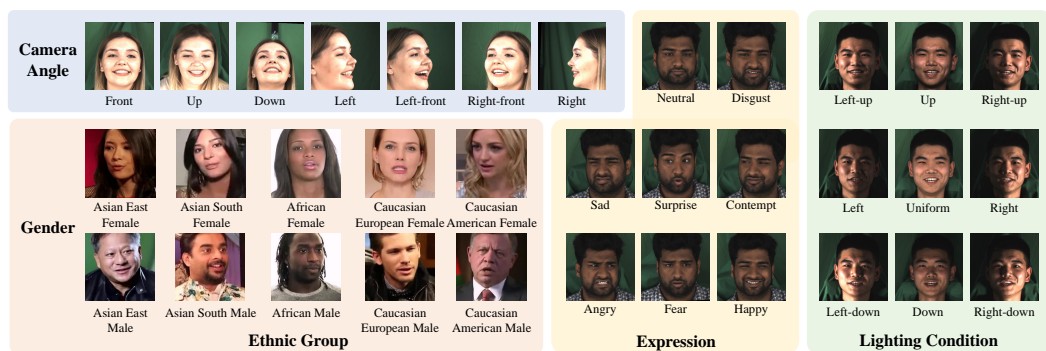

Figure 12: Overview of Attributes

data is a female. If the attribute bias of the detection algorithm is clear, proper improvements can be made to address the existing drawbacks. At the same time, it is also a timely reminder that inappropriate detection tasks with a basis should be avoided. Therefore, compared to evaluating the implicit bias of the deepfake category, the evaluation of explicit attribute bias is more important for practical implementation.

### B.5.1 VARIOUS ATTRIBUTES

We selected five attributes for evaluation, including Camera Angle, Gender, Ethnic Group, Expression, and Lighting Condition. Each Attribute contains 2-8 different categories.

**Camera angle**: This refers to the camera angle relative to the target when taking photos. The camera captures images from different angles. To evaluate whether the deepfake detection algorithm suffers detection bias for certain camera angles, the platform organizes the relevant data into 7 categories based on different camera angles, including Front, Up, Down, Left, Left front, Right front, and Right.

**Gender**: The platform also divides the data from public databases into two categories: Female and Male Then the platform can evaluate whether the detection algorithm has a bias in detecting gender.

**Ethnic group**: In this category, the platform divides data by ethnic group to evaluate whether detection algorithms have detection bias for ethnic groups. For example, when training detection algorithms, the database only has one race, such as white-skinned Caucasian Americans. During detection, the target face in the image to be tested may be an African with dark skin. The detection algorithm may not be able to perform accurate detection in this situation. Therefore, ethnic bias assessment is important. In this section, the platform divides data into the following 5 categories: Asian East, Asian South, African, Caucasian Europe, and Caucasian American.

**Expression**: Many facial data in public databases are dialogue scenes, so character expressions are related to conversations, and there are few special expressions like contempt. However, if the target face is manipulated into a special expression, it is difficult for the detection algorithm to perform detection accurately. If this kind of fake image spreads on the Internet, social stability will be seriously endangered. In this section, we divide data into the following 8 categories: Neutral, Distust, Sad, Surprise, Contempt, Angry, Fear, and Happy.

**Lighting Condition**: Most facial images in databases are collected under sufficient lighting conditions. but in reality, lighting conditions may be more complex. For example, in the "Right" image of the lighting condition in Figure 12, the left half of the face is almost in darkness. In this case, detection algorithms that are not trained for multiple lighting conditions may misclassify the real face as a fake face. In order to evaluate whether a detection algorithm suffers bias towards lighting conditions, the platform divides the data into the following 9 subcategories: Left up, Up, Right up, Left, Uniform, Right, Left down, Down, Right down

### B.5.2 Evaluation Data Preparation

In this section, the platform mainly extracts data from FakeAVCeleb and DeeperForensics-1.0 databases. Because these 2 databases have detailed data annotations for the 5 attributes mentioned above, data preparation is more convenient than other databases. There are 3 attributes whose data comes from DeeperForensics 1.0, namely Camera Angle, Expression, and Lighting Condition. Please note that all the data corresponding to these 3 attributes are real faces and there is no original fake face. The platform manipulates real faces through the manipulation algorithms in Section B.2.2. Then use the data before and after manipulation as real and fake face pairs for evaluation.

The data for the other 2 attributes comes from FakeAVCeleb, namely Gender and Ethnic Group. These 2 attributes have original real and fake faces, which can be directly used for evaluation. In order to enhance the generalization of this part, the platform also manipulates the real faces corresponding to these 2 attributes through the fake data generation pipeline mentioned in Section B.2.2. Then, resulting in a fake face with richer forgery traces.

### B.5.3 Fairness and Effectiveness Evaluation

This evaluation helps identify biases and weaknesses of the detection algorithm concerning specific attributes. Consequently, strategies to mitigate these biases, such as data balancing and attribute-aware training, can be proposed and validated.

## B.6 Forgery Localization Accuracy Evaluation

This section mainly evaluates the forgery localization ability of detection algorithms. In addition to distinguishing between fake and real images, deepfake detection algorithms will increasingly need to locate the manipulated regions in an image and the forged segments in a video in the future.

### B.6.1 Annotated Data for Localization

The platform uses data from the FaceForensics++ database to evaluate the manipulation region localization. The FaceForensics++ database includes annotation of the forged regions of faces. The platform uses Lav-DF database to evaluate video manipulation segment localization. The Lav-DF database contains detailed annotations of forged segments, with some videos even containing multiple forged segments, providing a robust test of the detection algorithm's ability to locate forged segments

### B.6.2 Localization Accuracy Metrics

This evaluation uses specialized metrics to quantify the algorithm's manipulation localization performance, including IoU (Intersection over Union), pixel accuracy, AP@IoU threshold (Average Precision at a specific IoU threshold), and AR@IoU threshold (Average Recall at a specific IoU threshold). Through this evaluation, the platform can analyze the localization ability of detection algorithms under different manipulation techniques, forgery region and shapes.

## C Experiments and Results

In this section, we introduce the experimental setup, evaluation results of each evaluation category, and insights behind the results.

## C.1 Experimental Setup

Experimental setup includes three parts: 1.Detection Algorithms; 2.Databases and Preprocessing; and 3.Evaluation Data Sampling.

**Detection Algorithms**: In this experiment, we evaluated 12 deepfake detection algorithms, including Xception (Rossler et al., 2019), SRM (Luo et al., 2021), SBI (Shiohara & Yamasaki, 2022), DSP-FWA (Li & Lyu, 2018), Multiple-Attention (Zhao et al., 2021), Efficientnetb4 (Tan & Le, 2019), MesoInception (Afchar et al., 2018), MesoNet (Afchar et al., 2018), UCF (Yan et al., 2023a),

CNN-Aug (Wang et al., 2020), ClassNSeg (Nguyen et al., 2019a), and BA-TFD+ (Cai et al., 2023). Among them, the first 10 are common detection algorithms which just discriminate real and fake images, while the last two are algorithms for detecting forgery regions and forgery segments, respectively.

**Databases and Preprocessing**: The platform implements 11 popular public databases, including FaceForensics++ (Rossler et al., 2019), Celeb-DF-v1 (Li et al., 2020b), Celeb-DF-v2 (Li et al., 2020b), FakeAVCeleb(Khalid et al., 2021), DeeperForensics-1.0 (Jiang et al., 2020), DFFD (Dang et al., 2020), DFDC (Dolhansky et al., 2020), CelebA (Liu et al., 2015), VGGface2 (Cao et al., 2018), VidTIMIT (Sanderson & Lovell, 2009), and Lav-DF (Cai et al., 2023). These databases cover data with labels of all four deepfake categories. In addition, it contains special data with labels of the forgery region, the forgery segment, and different attributes. The platform used the deployed deepfake manipulation algorithms to generate fake data following the pipeline described in Section B.2.2, resulting in over 5 million fake images for detector evaluation. The platform applied common image distortions to the data from the public database. Each real and fake image undergoes nine types of distortions. The platform also used adversarial attack algorithms to reconstruct and add perturbations to fake images. A detailed description of the data will be provided in each evaluation category.

**Evaluation Data Sampling**: The platform includes six evaluation categories, 27 evaluation subcategories, and over 40 sub-tasks. To ensure evaluation efficiency, we set 10,000 test samples for each sub-task, with 5,000 being real samples and 5,000 being fake samples. Additionally, to avoid test data bias caused by differences in the number of samples across databases, we sampled an equal number of data from each database to form the evaluation set. If the quantity is not an integer, it will be rounded up to the nearest integer. For example, in the Attribute Bias Assessment evaluation, the Happy sub-task under the Expression category consists of 5,000 forged data provided by 7 forgery algorithms, with 715 fake images randomly sampled from each algorithm.

## C.2 Benchmark Performance Evaluation

This experiment is to evaluate the benchmark performance of detection algorithms on public datasets.

### C.2.1 Databases and Preprocessing

This section utilizes seven databases, including Celeb-DF-V1, Celeb-DF-V2, DeeperForensics-1.0, FakeAVCeleb, FaceForensics++, DFFD and DFDC. These databases cover all four types of deepfake and have been preprocessed according to the unified pipeline described in Section B.1.2. Considering evaluation efficiency and avoiding data imbalance, the platform samples data according to the Evaluation Data Sampling method. All detection algorithms are evaluated through the same evaluation dataset.

### C.2.2 Evaluated Deepfake Detection Algorithms

The platform assesses 10 detection algorithms, including Xception (Rossler et al., 2019), SRM (Luo et al., 2021), SBI (Shiohara & Yamasaki, 2022), DSP-FWA (Li & Lyu, 2018), Multiple-Attention (Zhao et al., 2021), Efficientnetb4 (Tan & Le, 2019), MesoInception (Afchar et al., 2018), MesoNet (Afchar et al., 2018), UCF (Yan et al., 2023a), and CNN-Aug (Wang et al., 2020). To simulate the most realistic evaluation scenarios, we use the pre-trained models for each algorithm.

### C.2.3 Evaluation Results

We use the evaluation metrics mentioned in Section B.1.3. We present the results for Accuracy, AUC, F1-score, and EER. Additional evaluation metrics such as Precision can be found in the Appendix G.0.1.

Figure 6 shows the performance of 10 detectors. Overall, the accuracy of the detection algorithms is generally low. The highest accuracy achieved was 83.43%, while the lowest was only 30.4%. In Figure 6 (d), the F1-score for MesoNet and the Multi-Attention algorithm are very low because their recall values are particularly low, with maximum Recalls of only 0.16% and 1.08%, respectively.

Table 4: The detailed list of generated fake images.

| Index | Category | Sum | Algorithm | Number |
|---|---|---|---|---|
| 1 | | | FaceShifter | 596,416 |
| 2 | Face Swapping | 1,874,051 | MobileFaceSwap | 670,682 |
| 3 | | | FaceDancer | 606,953 |
| 4 | Facial Reenactment | 1,517,224 | HyperReenact | 941,082 |
| 5 | | | DGFR | 576,142 |
| 6 | Facial Attribute Manipulation | 1,635,821 | StarGAN2 | 420,000 |
| 7 | | | STGAN | 1,215,821 |
| 8 | | | ProGAN | 100,000 |
| 9 | Entire Face Synthesis | 540,000 | StyleGAN2 | 100,000 |
| 10 | | | StyleGAN3 | 100,000 |
| 11 | | | StyleGANV | 240,000 |
| 12 | | | Stable Diffusion | 120,552 |
| 13 | Text-to-Image | 307,979 | Mini-DALLE | 82,982 |
| 14 | | | LDM | 4,445 |
| 15 | | | Mini-DALLE3 | 100,000 |
| 16 | Image-to-Video | 101,070 | Stable Video Diffusion | 101,070 |

Half of the detection algorithms have an accuracy around 50%, regardless of the deepfake type of evaluation. The other half of the detectors show significant accuracy variations depending on the deepfake type. For instance, UCF achieved the highest AUC, Accuracy, F1-score, and the lowest EER in the Facial Reenactment category compared to the other three deepfake types. This indicates that UCF is most effective in detecting fake data in the Facial Reenactment category.

These experimental results reveal the basic detection performance of the algorithms and the specific forgery categories each detector excels at distinguishing. For other evaluation metrics results, please refer to Appendix G.0.1.

## C.3 FORGERY ALGORITHM GENERALIZATION ASSESSMENT RESULTS

This assessment is used to evaluate the generalization ability of detectors to unknown deepfake algorithms and data domains.

### C.3.1 DATABASES AND PREPROCESSING

The platform utilizes various deepfake algorithms to generate rich forged samples from real images/videos in multiple public databases. The platform configured 16 deepfake algorithms, covering various deepfake categories. Among them, FaceShifter (Li et al., 2019), FaceDancer (Rosberg et al., 2023), and MobileFaceSwap (Xu et al., 2022) belong to the Face Swapping , while DGFR (Hsu et al., 2022) and HyperReenact (Bounareli et al., 2023) belong to the Facial Reenactment. Additionally, STGAN (Liu et al., 2019) and StarGAN-V2 (Choi et al., 2020) belong to the Facial Attribute Manipulation. StyleGAN2 (Karras et al., 2020), StyleGAN3 (Karras et al., 2021), ProGAN (Karras et al., 2017), and StyleGAN-V (Skorokhodov et al., 2022) belong to the Entire Face Synthesis. LDM (Rombach et al., 2022b), Stable Diffusion (Rombach et al., 2022a), DALLE-mini(Dayma et al., 2021), and DALLE3-mini (Zeqiang et al., 2023) belong to Text-to-Image. Stable-Video-Diffusion (Blattmann et al., 2023) belongs to Image-to-Video.

Through the generation pipeline mentioned in Section B.2.2 and FFmpeg, the platform processed the data into a standard PNG image. This resulted in a total of 5,976,145 fake images. Table 4 provides detailed information on the fake images. Similarly, to ensure evaluation efficiency and fairness, the platform sampled the data.

### C.3.2 EVALUATION RESULTS

The detection algorithms evaluated in this section are consistent with those evaluated in Section C.2.2. Figure 7 shows the histogram results of the evaluation. Overall, the accuracy of the detectors remains generally low, and most detectors achieve an accuracy around 50%. The highest and lowest accuracies are 80.42% and 36.52%, respectively. Only EfficientNetB4 and UCF performed relatively well compared to the other eight detectors. Additionally, the Large Multimodal Model is the second most effectively detected category by them. The detection algorithms show

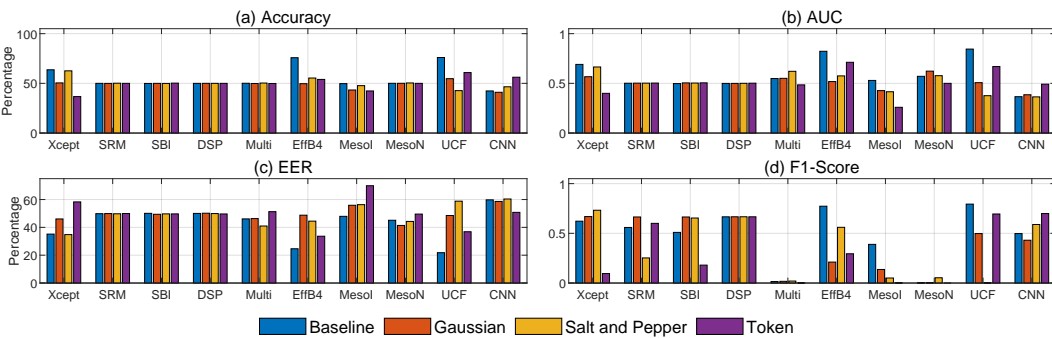

Figure 13: Image Distortion Robustness Evaluation (Noise): comparison of Accuracy, AUC, EER, and F1-Score among 10 deepfake detection algorithms.

different preferences for five deepfake categories. For instance, UCF achieved the highest accuracy, AUC, F1-score, and the lowest EER in the Facial Reenactment category. Its AUC was 0.2677 higher than its performance in the least favorable category, Entire Face Synthesis.

These experimental results highlight the detection generalization capabilities of each detector for different forgery categories. For other evaluation metrics results, please refer to Appendix G.1.

### C.4 IMAGE DISTORTION ROBUSTNESS ASSESSMENT RESULTS

This section is used to evaluate whether the detectors are susceptible to common image distortions.

#### C.4.1 DATABASES AND PREPROCESSING

To clearly reflect the impact of common image distortions, the platform uses the Benchmark Evaluation Performance data as the original data for processing, with the corresponding detection results serving as the baseline.

This part includes 9 types of distortions: Compression, Brightness, Contrast, Flip, Rotation, Color, Sharpness, Blur, and Noise. Each distortion category includes 2-8 degrees. For example, Blur includes Gaussian, Mean, and Median blurs. All distortion outputs for a single image can be referred to Figure 11.

The platform applies each of these distortions to the original data and saves the results as images. Except for Compression, which is saved in JPG format, all other images are saved in PNG format. Since the original data is obtained through uniform sampling, there is no concern about sample imbalance.

#### C.4.2 EVALUATION RESULTS

We present the results for four types of distortions: Compression, Noise, Blur, and Rotation. For better comparison, the results for the four deepfake categories are averaged in this section. For example, the accuracy result of 0.85 in Compression corresponds to the average accuracy of the four deepfake categories at a compression ratio of 0.85.

**Compression**: Figure 8 shows the trend of evaluation metrics for 10 detectors as the compression ratio decreases. Overall, detectors with initially higher accuracy show a gradual decrease in accuracy as the compression ratio decreases. However, for detectors with initially low accuracy, the trend is not as apparent. EfficientNetB4 and UCF remain the strongest detectors, consistently achieving the top two positions across the four metrics under various compression ratios.

**Noise**: Figure 13 shows the impact of noise on the 10 detectors. Overall, detectors with an initial accuracy around 50% are less affected by noise, while those with higher initial accuracy are significantly impacted. For example, EfficientNetB4's average accuracy was 75.795%, but it dropped by 26.14%, 20.43%, and 21.83% respectively under three types of noise, falling to around 50%.

**Blur**: Figure 14 shows the impact of blur on the 10 detectors. Overall, for detectors with initially

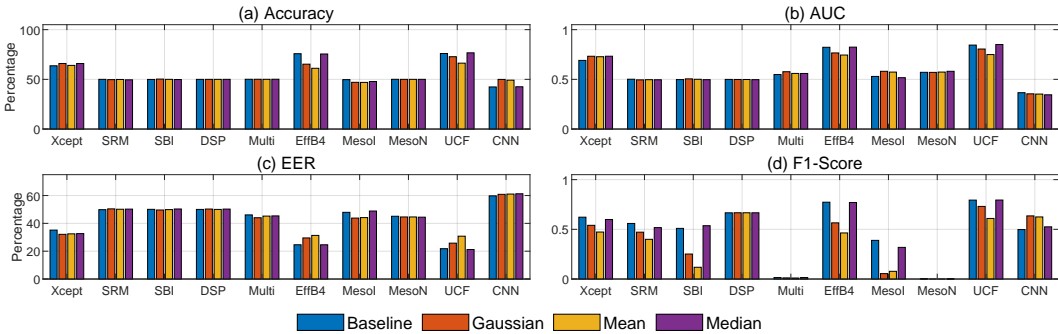

Figure 14: Image Distortion Robustness Evaluation (Blur): comparison of Accuracy, AUC, EER, and F1-Score among 10 deepfake detection algorithms.

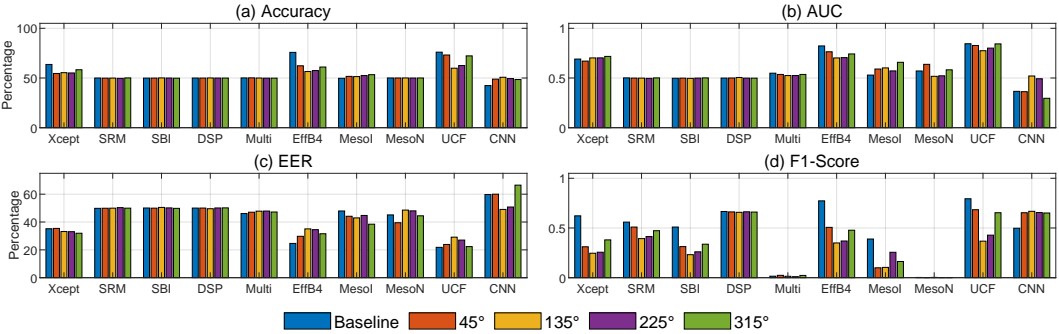

Figure 15: Image Distortion Robustness Evaluation (Rotation): comparison of Accuracy, AUC, EER, and F1-Score among 10 deepfake detection algorithms.

high accuracy, Gaussian and Mean blur methods significantly impact the results, whereas the Median blur method has a smaller effect. The impact of blur on detectors is generally less than that of noise.

**Rotation**: Figure 15 shows the impact of rotation on the 10 detectors. Overall, for detectors with initially high accuracy, rotation at 135 and 225 degrees causes a significant drop in accuracy, while rotations at 45 and 315 degrees cause a smaller drop. This may be because the face is almost upside down at 135 and 225 degrees, deviating significantly from the usual facial orientation.

These evaluation results highlight how common image distortions affect the performance of various detection algorithms, providing insights into their robustness and potential areas for improvement. For results of other distortions, please refer to Appendix G.2.

## C.5 ADVERSARIAL ATTACK RESILIENCE EVALUATION

This section evaluates whether the detectors are capable of resisting adversarial attacks.

### C.5.1 DATABASES AND PREPROCESSING

To facilitate comparison of performance changes before and after adversarial attacks, this section uses the data processed for the Benchmark Performance Evaluation as the original data, with the corresponding detection results serving as the baseline.

For **Images Reconstruction**, the platform uses the GANprintR algorithm. This algorithm is primarily used to remove GAN "fingerprints" from synthetic fake images. Therefore, we focus on the Entire Face Synthesis category for data processing and evaluation. Real images remain unchanged, while synthetic fake images undergo image reconstruction. The reconstructed data, along with the real images, form the evaluation dataset.

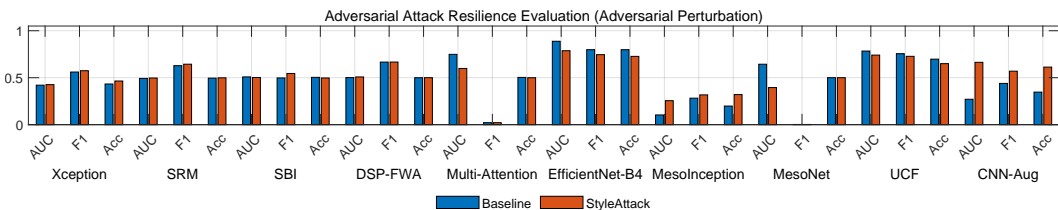

Figure 16: Adversarial Attack Resilience Evaluation (Adversarial Perturbation): comparison of AUC, F1-Score, and Acc among 10 deepfake detection algorithms.

For **Adversarial Perturbation**, the platform employs the StyleAttack algorithm. This algorithm searches for adversarial points in the latent space of a generative model to generate anti-forensic fake face images. Since this attack mainly targets GAN-related algorithms, the real images used are the same as those in the Entire Face Synthesis category. The platform uses the StyleAttack algorithm to generate a large number of fake synthetic images, forming the required evaluation dataset.

### C.5.2 EVALUATION RESULTS

Since both attack algorithms target GANs, we use the detection results of the Entire Face Synthesis category from the Benchmark Performance Evaluation as the baseline for comparison before and after the attack.

**GANprintR**: Figure 9 shows the experimental results of GANprintR, including AUC, F1-score, and accuracy. To bring all three evaluation metrics to the same value range, percentages are converted to decimal form. Overall, GANprintR does not deceive all detection algorithms. Compared to the baseline, the accuracy of Xception and UCF decreases when facing GANprintR-processed data, indicating that these detectors failed to resist GANprintR. On the other hand, the accuracy of EfficientNetB4 increases when facing GANprintR-processed data, indicating that EfficientNetB4 successfully resisted this attack.

**StyleAttack**: Figure 16 shows the experimental results of StyleAttack. Overall, StyleAttack affects different detectors to varying degrees. For detectors with initially high accuracy, StyleAttack reduces their accuracy, indicating the effectiveness of the attack. For detectors with initially low accuracy, StyleAttack actually increases their accuracy.

This evaluation verifies whether detectors have the capability to resist adversarial attacks. As research on evading detection continues to mature, this will become an increasingly important evaluation in the future, despite the current limited research in this area. For other evaluation metrics results, please refer to Appendix G.3.

### C.6 FORGERY LOCALIZATION ACCURACY EVALUATION

This section aims to fulfill specific evaluation requirements for forgery localization, focusing on forgery regions and forgery segments.

### C.6.1 DATABASES AND PREPROCESSING

For **forgery region evaluation**, the platform utilizes data from FaceForensics++ with forgery region masks as the assessment dataset. The masks indicate the positions of forged pixels in a given image if it contains a fake face; otherwise, there are no forged pixels marked. The platform randomly selects 5000 real images and 5000 fake images from the FaceForensics++ database to form the evaluation dataset for this evaluation.

For **forgery segment evaluation**, the platform employs the Lav-DF database as the assessment data. This database contains both unforged videos and multiple forged video segments. Each video segment is accompanied by detailed label information, including the number of forged segments and the start and end timestamps of each forgery. The platform randomly selects 5000 real videos and 5000 fake videos from the database to constitute the evaluation dataset for this category.

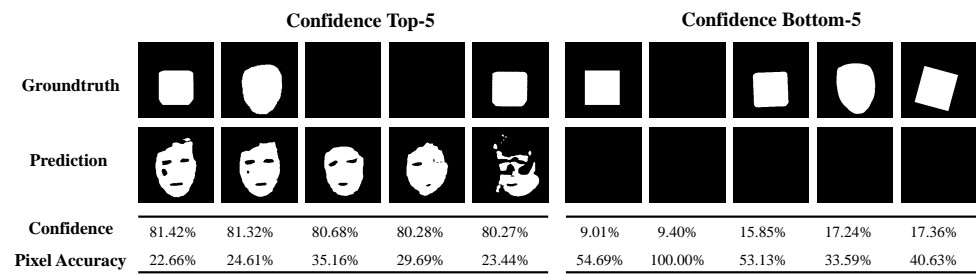

Figure 17: Example of Forgery Region Localization.

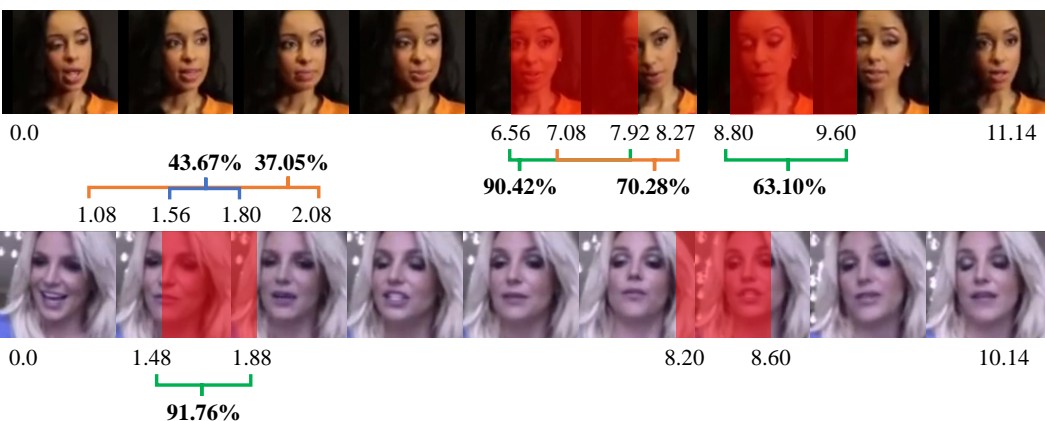

Figure 18: Example of Video Forgery Segment Localization.

### C.6.2 EVALUATION RESULTS

This section does not compare with previous baselines, but instead, evaluates the algorithms using the prepared evaluation datasets.

For **forgery region evaluation**, the platform assesses the ClassNSeg algorithm.

Table 2 presents the evaluation metrics of the ClassNSeg algorithm. While the recall reaches 86.95%, the accuracy and precision are relatively low. To gain a deeper understanding of the algorithm's localization ability for forgery regions, the platform compares the top 5 highest-confidence and the bottom 5 lowest-confidence predictions with the ground truth, and calculates the pixel accuracy accordingly. Figure 17 shows the predicted forgery regions. Among the five highest-confidence predictions, three are correctly identified, while two real images are falsely classified as fake. Only one of the five lowest-confidence predictions is correctly predicted.

For **forgery segment evaluation**, the platform evaluates the BA-TFD+ algorithm. Table 3 shows the average precision (AP) at different IoU thresholds and the average recall (AR) for different numbers of proposals with IoU thresholds ranging from 0.5 to 0.95 with a step size of 0.05. At an IoU of 0.5, the AP reaches 97.68%. When the number of proposals is 100, the AR exceeds 80%.

Figure 18 clearly shows the localization capability of the proposed detection algorithm. The two forged videos each contain two forged segments, marked as red regions in the videos. The figure highlights the three segments with the highest confidence scores predicted as fake. Green, orange, and blue lines represent the predicted forged segments, accompanied by their corresponding confidence scores. It can be observed from the figure that all the forged segments in the first video have been successfully localized, while only one forged segment in the second video has been localized.

This evaluation fulfills the specific requirements for assessing the forgery location capabilities of detection algorithms.

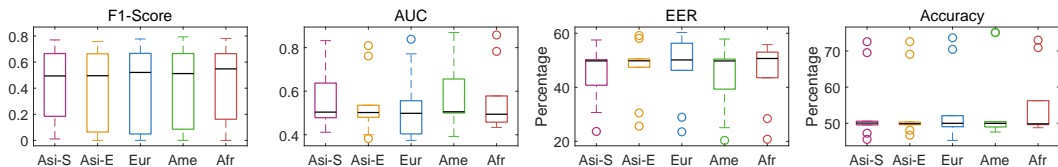

Figure 19: Attribute Bias Evaluation (Ethnic Group): comparison of F1-Score, AUC, EER and Accuracy among 10 deepfake detection algorithms.

## C.7 SCENE ATTRIBUTE BIAS ASSESSMENT

This section primarily focuses on assessing the existence of attribute bias in the detectors.

### C.7.1 DATABASES AND PREPROCESSING

The platform has selected five attributes for evaluation, including Gender, Ethnic Group, Camera Angle, Expression, and Lighting Condition. Among them, Gender is categorized into Male and Female. Ethnic Group comprises Asian East, Asian South, African, Caucasian European, and Caucasian American. Camera Angle encompasses seven categories such as Front, Up, Down. Expression includes eight categories like Happy, Sad, Surprise. Lighting Condition covers nine types like Left, Uniform, Right. Specific examples are provided in Figure 12.

The platform gathered these data from public databases, and for those data lacking corresponding fake images, we utilized the pipeline introduced in Section B.2.2 to generate fake images. Ultimately, a real-fake dataset was generated for the evaluation of each attribute.

### C.7.2 EVALUATION RESULTS

This section presents the experimental results for the Ethnic Group and Camera Angle attributes. For other results, please refer to Appendix G.4. To explore whether there exists a universal attribute bias in the current detectors, we plotted boxplots of the results from 10 detectors under different evaluation metrics.

Figure 19 displays the evaluation results for ethnic group. It can be observed from the figure that the black median lines corresponding to different ethnic groups are relatively similar. The four evaluation metrics reveal different biased categories. Therefore, overall, these detectors under test do not exhibit bias towards ethnic groups.

Figure 10 presents the evaluation results for Lighting Condition. It is evident from the figure that the black median lines corresponding to different lighting conditions vary significantly, indicating a clear bias. Analysis of the boxplot range reveals that the UP category has the highest F1-score, AUC, accuracy, and the lowest EER.

This evaluation can determine whether a detector exhibits bias towards a specific attribute. It allows for the timely identification of weaknesses and guides the optimization of detectors in a more comprehensive direction. For results of other attributes, please refer to Appendix G.4.

## D LIMITATIONS

The proposed Deepfake Detector Assessment Platform (DAP) offers a comprehensive evaluation framework for assessing the performance, generalization ability, robustness, security, localization precision, and fairness of deepfake detection algorithms. However, there are a few limitations that can be addressed in future work to further enhance the platform's capabilities and real-world applicability.

One limitation is that the platform currently focuses on image-based deepfake detection, while video-based deepfakes are becoming increasingly prevalent and pose unique challenges. Extending the platform to support video-based deepfake detection would require the incorporation of temporal information and the development of suitable evaluation metrics that consider the consistency and

coherence of detected forgeries across frames. This extension would enable the platform to provide a more comprehensive assessment of deepfake detection algorithms' performance in real-world scenarios.

Another limitation is related to the diversity and complexity of the datasets used for evaluation. While the platform incorporates a wide range of public and self-generated datasets, it could benefit from the inclusion of even more challenging datasets that cover extreme poses, occlusions, low-resolution images, and other factors that are commonly encountered in real-world settings. Evaluating detection algorithms on these challenging datasets would provide a more accurate assessment of their robustness and ability to handle diverse and complex cases.

Lastly, the computational efficiency of the platform could be improved to handle large-scale evaluations more effectively. As the number of deepfake generation and detection algorithms continues to grow, along with the size and complexity of datasets, optimizing the evaluation pipeline and leveraging parallel processing techniques would enable faster and more efficient assessments. This would facilitate the timely evaluation of new algorithms and the ability to keep pace with the rapidly evolving landscape of deepfake technologies.

## E  FUTURE WORK

The development of the Deepfake Detector Assessment Platform opens up several exciting avenues for future research and improvement. One key direction is to extend the platform to support video-based deepfake detection, as mentioned in the limitations section. This would involve incorporating temporal information, such as frame-level consistencies and inconsistencies, into the evaluation process. Furthermore, developing suitable evaluation metrics that consider the temporal aspects of video-based deepfakes would provide a more comprehensive assessment of detection algorithms' performance in this domain.

Another important future direction is to establish collaborations with the research community to ensure that the platform remains up-to-date with the latest advancements in deepfake generation and detection algorithms. By actively engaging with researchers and practitioners, the platform can incorporate state-of-the-art techniques and datasets, ensuring its continued relevance and effectiveness in the rapidly evolving field of deepfake detection. This collaborative approach would also facilitate the sharing of knowledge and best practices, fostering innovation and accelerating progress in this critical area.

Investigating the potential of integrating explainable AI techniques into the platform is another promising future direction. Explainable AI aims to provide insights into the decision-making process of machine learning models, making them more interpretable and trustworthy. By incorporating explainable AI techniques into the evaluation process, the platform could offer a deeper understanding of how deepfake detection algorithms arrive at their decisions, identifying the key features and patterns they rely on. This would not only enhance the interpretability of the detection algorithms but also facilitate the development of more robust and reliable models.

Exploring the use of transfer learning and few-shot learning approaches is another area of future work that could significantly improve the generalization ability of deepfake detection algorithms. Transfer learning involves leveraging knowledge gained from one task or domain to improve performance on another related task or domain, while few-shot learning aims to learn from a limited number of examples. By incorporating these approaches into the evaluation process, the platform could assess the ability of detection algorithms to adapt to new deepfake generation techniques and datasets with minimal retraining, which is crucial in real-world scenarios where labeled data may be scarce.

Finally, developing a user-friendly interface and visualization tools for the platform would greatly enhance its accessibility and usability for researchers, practitioners, and policymakers. An intuitive interface would allow users to easily configure and run evaluations, while interactive visualization tools would enable them to explore and analyze the results in a meaningful way. This would promote wider adoption of the platform and facilitate collaboration among stakeholders, ultimately contributing to the development of more effective and trustworthy deepfake detection solutions.

## F   BROADER SOCIETAL IMPACT

The development of a comprehensive Deepfake Detector Assessment Platform has far-reaching societal implications, as it directly addresses the growing concern over the malicious use of deepfakes and their potential to erode trust in digital media. By providing a standardized and rigorous evaluation framework, the platform enables the development of reliable and robust deepfake detection algorithms, which is crucial for combating the spread of misinformation, fraud, and other forms of manipulation.

In an era where deepfakes are becoming increasingly sophisticated and accessible, the ability to accurately detect and flag manipulated content is essential for maintaining the integrity of digital media. The platform's comprehensive evaluation approach, covering performance, generalization, robustness, security, localization, and fairness, ensures that detection algorithms are assessed under a wide range of realistic conditions. This helps to identify strengths and weaknesses of existing algorithms and guides the development of more effective and resilient detection methods. By facilitating the creation of trustworthy deepfake detection solutions, the platform contributes to safeguarding individuals, organizations, and society as a whole from the potential harm caused by malicious actors exploiting deepfake technologies.

Moreover, the platform's emphasis on assessing attribute bias and fairness in deepfake detection algorithms is a critical step towards promoting the development of more inclusive and unbiased systems. Algorithmic bias is a significant concern in many domains, as it can perpetuate and amplify existing societal biases and lead to discriminatory outcomes. By explicitly evaluating the performance of detection algorithms across different demographics, contexts, and attributes, the platform helps to identify and mitigate potential biases. This is essential for ensuring that deepfake detection technologies are deployed in a responsible and equitable manner, preventing the exacerbation of existing inequalities and promoting fairness in the digital realm.

The open-source nature of the platform and the accompanying benchmark results also have significant societal benefits. By making the platform and its findings publicly available, the research community can collaborate more effectively, sharing knowledge, insights, and best practices. This collaborative approach accelerates progress in deepfake detection research, enabling the development of more advanced and robust countermeasures. Furthermore, the transparency provided by the platform helps to build public trust in deepfake detection technologies, as the evaluation process and results are open to scrutiny and validation by the wider community.

In conclusion, the Deepfake Detector Assessment Platform has the potential to make a significant positive impact on society by promoting the development of reliable, fair, and trustworthy deepfake detection solutions. By providing a comprehensive and standardized evaluation framework, the platform contributes to safeguarding the integrity of digital media, protecting individuals and organizations from harm, and fostering a more resilient and equitable digital ecosystem.

## G   OTHER DETAILED EXPERIMENTAL RESULTS

In this section, we introduce detailed experimental results of various evaluations.

The analysis in the main text primarily used AUC, F1-score, accuracy, and EER as the evaluation metrics. Here, we present seven evaluation metrics, including Precision, Recall, and confidence difference. To save space, we use three histograms: (1) Accuracy, Precision, and Recall; (2) AUC and F1-score; (3) confidence difference and EER.

### G.0.1   BENCHMARK PERFORMANCE EVALUATION

Figure 20, Figure 21, and Figure 22 show the histograms of benchmark performance evaluation results for different evaluation metrics. The recall values are very low for Multi-Attention and MesoNet. As a result, the F1-score values are very low for both detectors.

Figure 23 and Figure 24 show the Precision-Recall Curve and ROC Curve among 10 deepfake detectors.

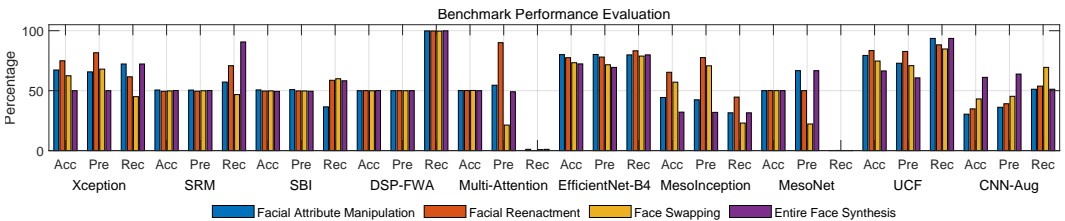

Figure 20: Benchmark Performance Evaluation: comparison of Accuracy, Precision, and Recall among 10 deepfake detection algorithms.

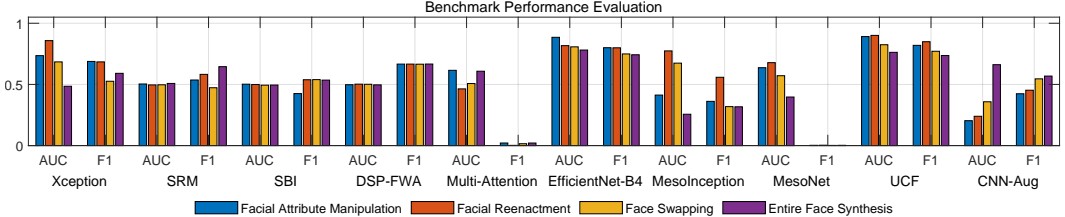

Figure 21: Benchmark Performance Evaluation: comparison of AUC and F1-Score among 10 deepfake detection algorithms.

### G.1 FORGERY ALGORITHM GENERALIZATION ASSESSMENT

Figure 25, Figure 26, and Figure 27 show the histograms of forgery algorithm generalization assessment results for different evaluation metrics.

Figure 28 and Figure 29 show the Precision-Recall Curve and ROC Curve among 10 deepfake detectors.

### G.2 IMAGE DISTORTION ROBUSTNESS ASSESSMENT

Figure 30, Figure 31, Figure 32, Figure 33, and Figure 34 show the histograms of Accuracy, AUC, EER, and F1-score among 10 deepfake detectors.

### G.3 ADVERSARIAL ATTACK RESILIENCE EVALUATION

Figure 35 and Figure 36 show the Adversarial Perturbation evaluation results of accuracy, precision, recall, EER, and confidence difference among 10 deepfake detection algorithms.

Figure 37 and Figure 38 show the Image Reconstruction evaluation results of accuracy, precision, recall, EER, and confidence difference among 10 deepfake detectors.

### G.4 ATTRIBUTE BIAS ASSESSMENT

Figure 39, Figure 40, and Figure 41 display the evaluation results for camera angle, expression and gender.

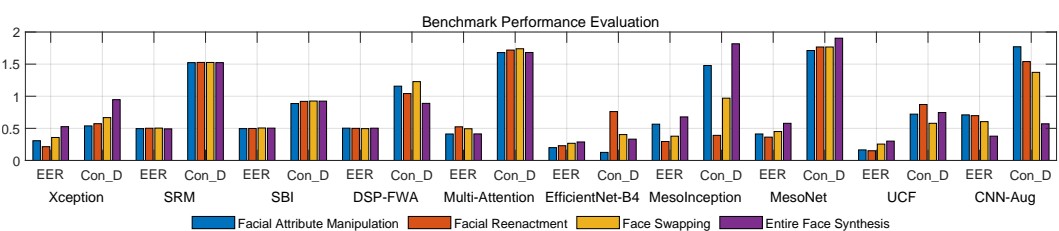

Figure 22: Benchmark Performance Evaluation: comparison of Confidence Difference and EER among 10 deepfake detection algorithms.

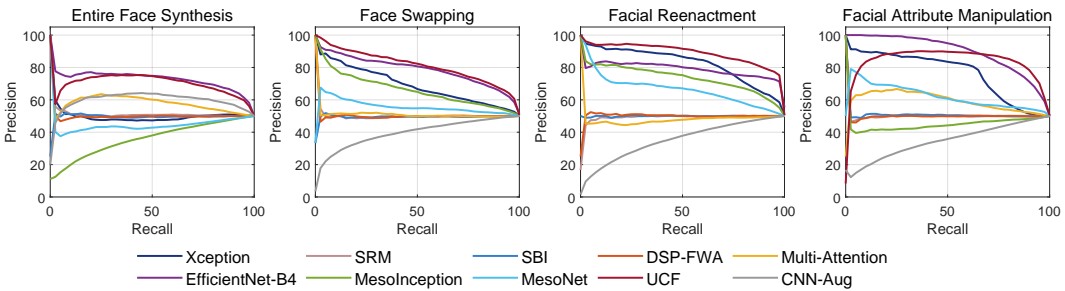

Figure 23: Benchmark Performance Evaluation: comparison of Precision-Recall Curve among 10 deepfake detection algorithms.

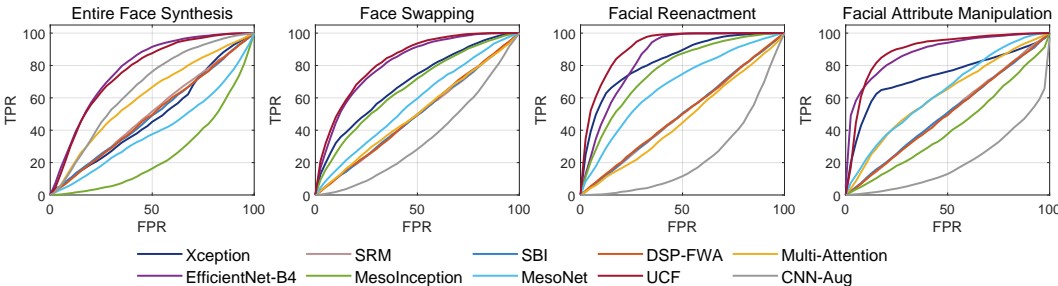

Figure 24: Benchmark Performance Evaluation: comparison of ROC Curve among 10 deepfake detection algorithms.

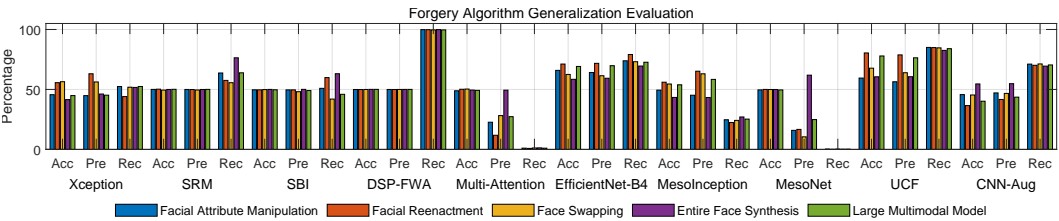

Figure 25: Forgery Algorithm Generalization Assessment: comparison of Accuracy, Precision, and Recall among 10 deepfake detection algorithms.

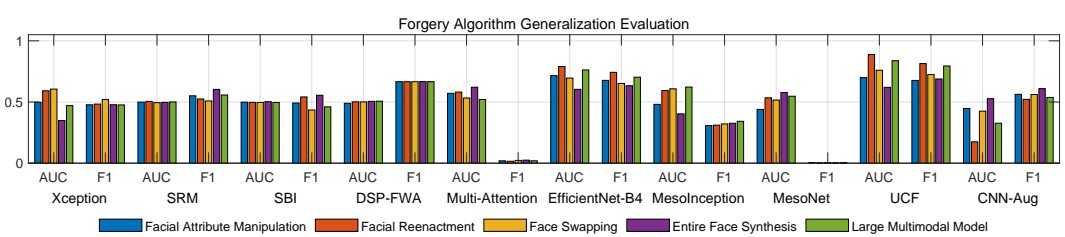

Figure 26: Forgery Algorithm Generalization Assessment: comparison of AUC and F1-Score among 10 deepfake detection algorithms.

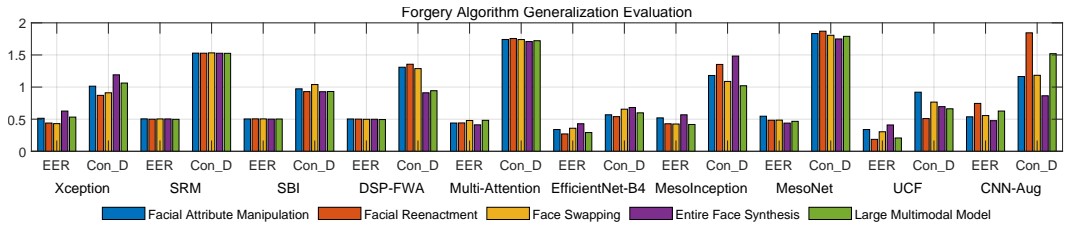

Figure 27: Forgery Algorithm Generalization Assessment: comparison of Confidence Difference and EER among 10 deepfake detection algorithms.

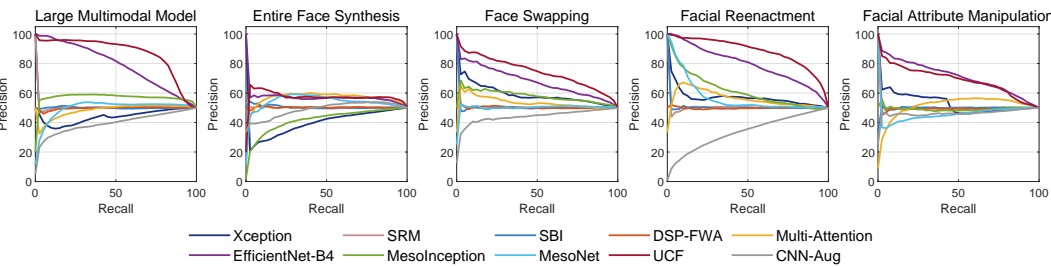

Figure 28: Forgery Algorithm Generalization Assessment: comparison of Precision-Recall Curve among 10 deepfake detection algorithms.

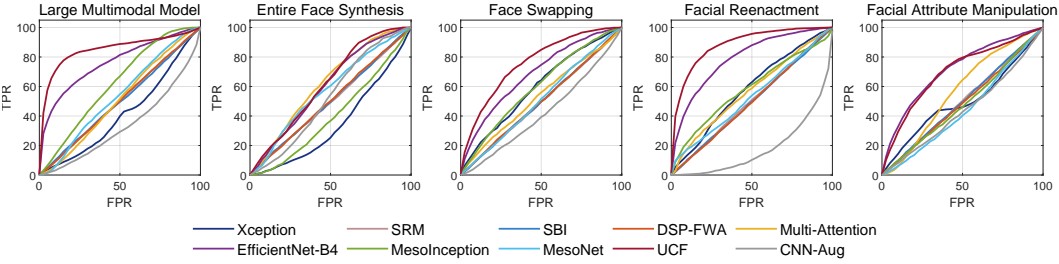

Figure 29: Forgery Algorithm Generalization Assessment: comparison of ROC Curve among 10 deepfake detection algorithms.

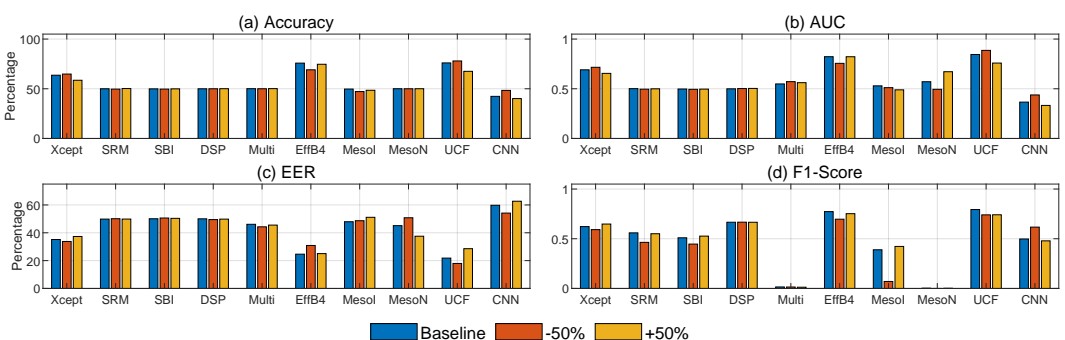

Figure 30: Image Distortion Robustness Evaluation (Brightness): comparison of Accuracy, AUC, EER, and F1-Score among 10 deepfake detection algorithms.

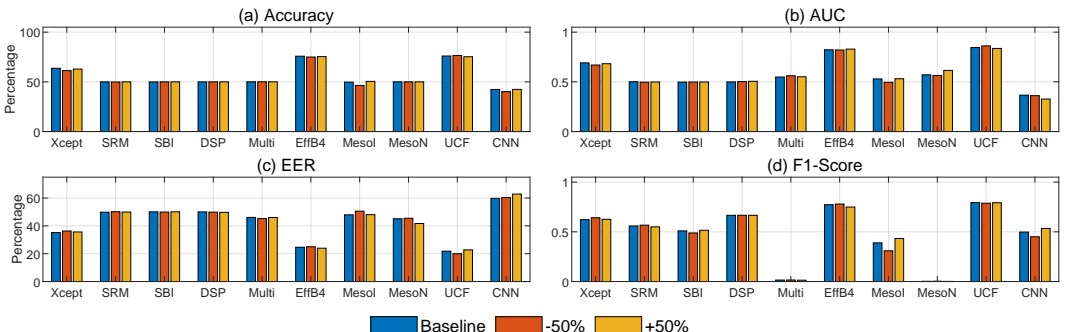

Figure 31: Image Distortion Robustness Evaluation (Color): comparison of Accuracy, AUC, EER, and F1-Score among 10 deepfake detection algorithms.

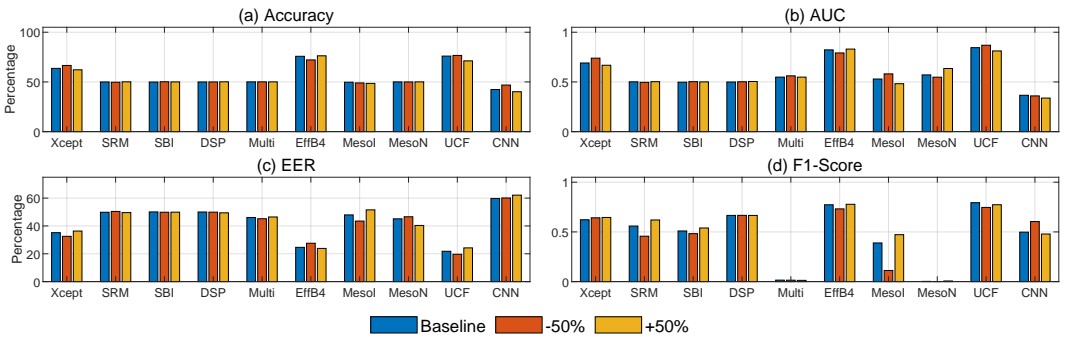

Figure 32: Image Distortion Robustness Evaluation (Contrast): comparison of Accuracy, AUC, EER, and F1-Score among 10 deepfake detection algorithms.

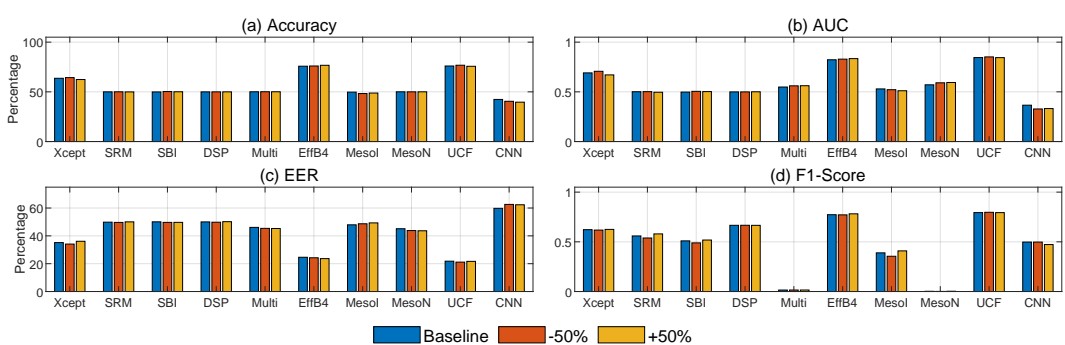

Figure 33: Image Distortion Robustness Evaluation (Sharpness): comparison of Accuracy, AUC, EER, and F1-Score among 10 deepfake detection algorithms.

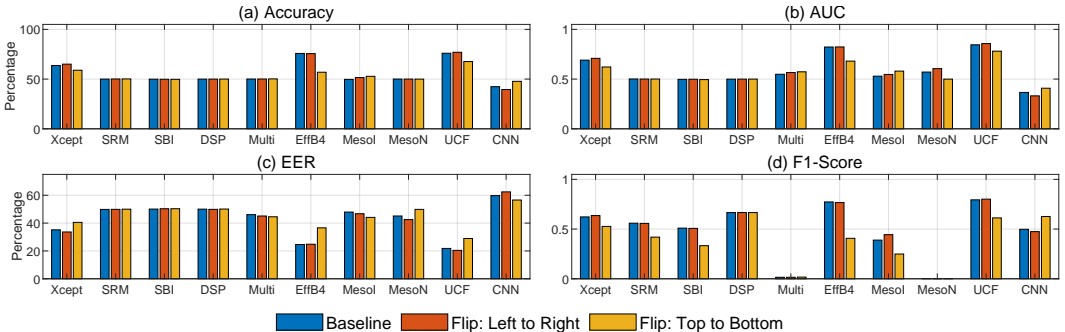

Figure 34: Image Distortion Robustness Evaluation (Flip): comparison of Accuracy, AUC, EER, and F1-Score among 10 deepfake detection algorithms.

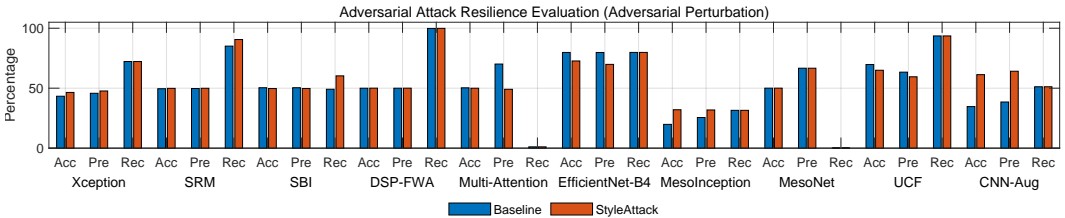

Figure 35: Adversarial Attack Resilience Evaluation: comparison of Accuracy, Precision, and Recall among 10 deepfake detection algorithms.

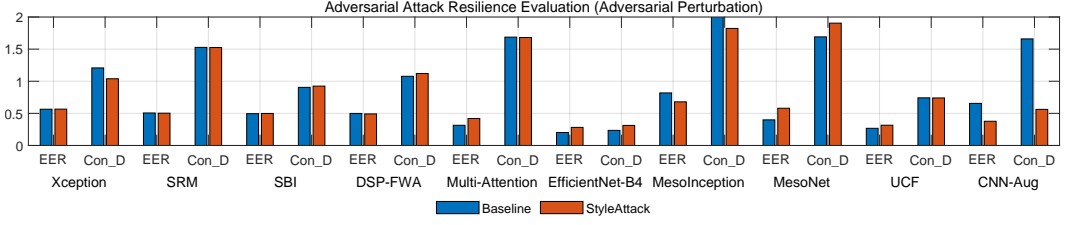

Figure 36: Adversarial Attack Resilience Evaluation (Adversarial Perturbation): comparison of Confidence Difference and EER among 10 deepfake detection algorithms.

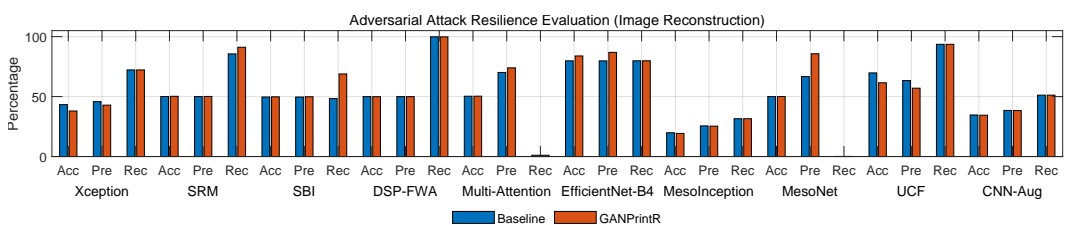

Figure 37: Adversarial Attack Resilience Evaluation (Image Reconstruction): comparison of Accuracy, Precision, and Recall among 10 deepfake detection algorithms.

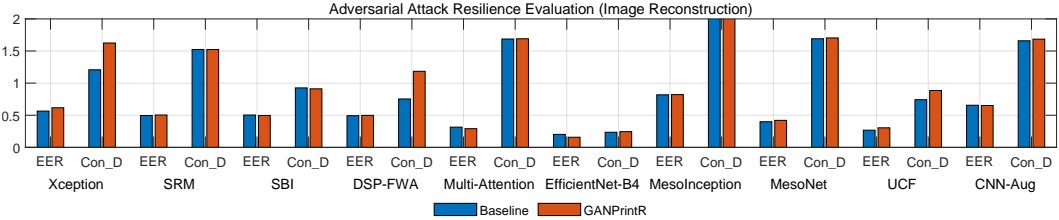

Figure 38: Adversarial Attack Resilience Evaluation (Image Reconstruction): comparison of Confidence Difference and EER among 10 deepfake detection algorithms.

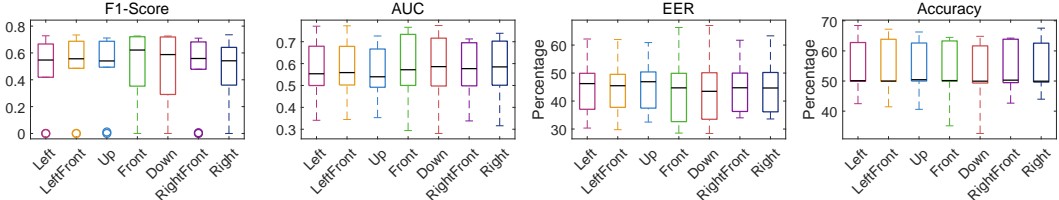

Figure 39: Attribute Bias Evaluation (Camera Angle): comparison of F1-Score, AUC, EER and Accuracy.

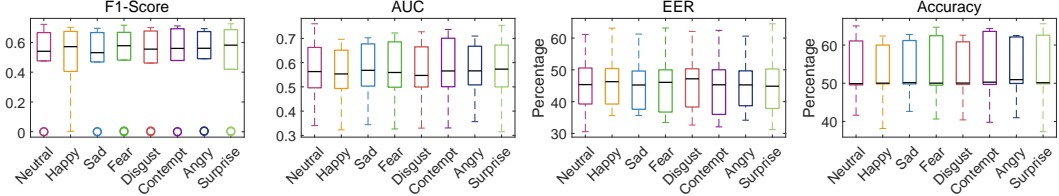

Figure 40: Attribute Bias Evaluation (Expression): comparison of F1-Score, AUC, EER and Accuracy.

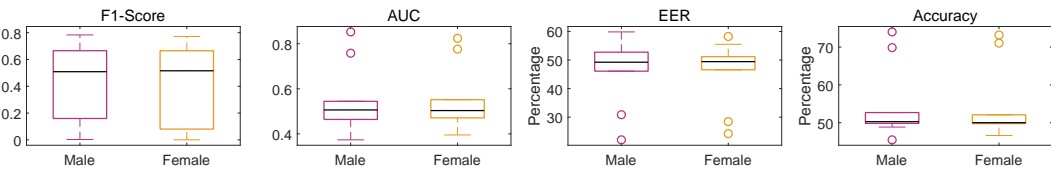

Figure 41: Attribute Bias Evaluation (Gender): comparison of F1-Score, AUC, EER and Accuracy.

