# OpenReview forum: "A Comprehensive Deepfake Detector Assessment Platform"
_ICLR.cc/2025/Conference — Submitted to ICLR 2025_

### Official Review · Reviewer_QXSL · 2024-11-03

**Soundness:** 2
**Presentation:** 2
**Contribution:** 2
**Rating:** 5
**Confidence:** 4

**Summary:**

This paper presents a comprehensive Deepfake Detector Assessment Platform (DAP) designed to evaluate the performance, generalization capability, robustness, security, localization accuracy, and fairness of deepfake detection algorithms. The platform covers six key dimensions: benchmark performance, forgery algorithm generalization, image distortion robustness, resilience against attacks, attribute bias, and forgery localization accuracy. Through extensive experiments on multiple public and self-built databases, the framework provides researchers and practitioners with a standardized and rigorous evaluation tool to develop and assess new approaches in the field.

**Strengths:**

1. A comprehensive evaluation framework covering multiple key dimensions of deep forgery detection is provided.
2. The experiments are rigorously designed, using multiple public and self-built databases, as well as a large number of self-generated fake images.
3. Adversarial attacks and attribute bias evaluation were introduced, which are important considerations for practical applications.

**Weaknesses:**

1. The experimental settings are not reasonable in some scenarios.
2. The evaluation experiments are not sufficient.
3. Lacking meaningful conclusions and findings.
4. Video-based deep forgery detection is equally important but has not yet been addressed.

**Questions:**

1. Authors choose forgery location precision as the critical dimension. Not all forgery detection models can localize the forgery region, so it is not reasonable to select it as the unified performance metric.
2. It is not necessary to show the image processing in Fig 3 and Fig 4.
3. In the generalization evaluation, these detection algorithms are not newly published, and only UCF is designed for generalized forgery tasks. Thus, this sufficient experiment can’t draw meaningful conclusions and findings.
3. In the adversarial perturbation experiments, authors mentioned they chose the StyleAttack algorithm as the attack method. But can it be regarded as the representative attack algorithm? Please give the reasons.
4. For the attribute bias evaluation, the authors summarize ten detection algorithms with different metrics. However, it lacks valuable conclusions and analysis. How does it affect further detector design?
5. The Large Multimodal models are considered as the generation methods. What changes and differences have been brought about by this strategy? Unfortunately, the author does not discuss this.

Overall, the paper seems more like a research report, lacking of In-depth experimental analysis and theoretical hypothesis.

---

### Official Review · Reviewer_bffT · 2024-11-03

**Soundness:** 3
**Presentation:** 4
**Contribution:** 3
**Rating:** 3
**Confidence:** 4

**Summary:**

Glad to review the paper.

This paper proposes a comprehensive deepfake detector assessment platform, which could provide evaluating results from multiple aspects.
The platform includes organized datasets, detectors, adversarial algorithms, etc., to support future work.

In general, I believe the work is referenceable to related-domain researchers.

**Strengths:**

This work considers assessing deepfake detectors from multiple aspects, e.g., generalization, distortion robustness, adversarial attack resilience, forgery localization, and attribute, the range of the work is wide enough.

The evaluation results in this work are large enough, in addition, the authors provide a tool (platform) to support future work.

**Weaknesses:**

I am concerned that the evaluation experiments conducted by the authors could not support the goals of this work.

(1) Regarding baseline detectors, the utilized 10 detectors are not well-categorized and are not with enough explanations, it is not clear why the 10 are selected, are they state-of-the-art or covering different types? , e.g., FaceX-ray (Li et al., 2020a) and Frequency-
aware method (Tan et al., 2024) are not evaluated, and whether the selected detectors cover : data-driven, spatial artifact-based, and frequency artifact-based types. In fact, in the related work part, although some detectors are listed, the state-of-the-art (or some typical) detectors are not introduced. I doubt the selected detectors would cause biased conclusions.

(2) Regarding innovation, the authors claim that the platform is different from existing work from the aspects of attribute assessment, adversarial attack, and forgery localization. However, only two adversarial attacks (GANprintR is the main one) are implemented on (only) two baseline detectors, and forgery localization is evaluated based on one detector only. First, the experimental conclusions would be biased based on limited attempts, such experimental settings should be explained in detail. Second, whether more adversarial attacks or forgery localization algorithms could be integrated into the platform or implemented into more baseline detectors?

(3) Regarding findings, although multiple assessments are conducted in the work, it will be useful to summarize some findings from these aspects of evaluations, e.g., why the detectors are not generalizable or vulnerable. which aspects should these detectors improve from? any future directions for these detectors?  A "Findings and Future Directions" section should be included that summarizes the findings from the results across every evaluation aspect and provides concrete recommendations for improving deepfake detection algorithms (based on the findings).

**Questions:**

My major concerns are summarized in the weakness part, questions from three aspects are expected to be responded.

I could consider changing the score if the responses are convincing enough, or any of my misunderstandings exist.

---

### Official Review · Reviewer_M56J · 2024-11-04

**Soundness:** 2
**Presentation:** 1
**Contribution:** 2
**Rating:** 3
**Confidence:** 5

**Summary:**

This paper proposes a platform for assessing deepfake detection algorithms in terms of their performance, generalizability, robustness, security, localization precision, and fairness. Extensive experimental evaluations were conducted on public and author-generated datasets, showing limited performance of the existing deepfake detectors in the literature. The paper aims to offer insights into the strengths and limitations of the existing deepfake detectors. The code and data are publicly released.

**Strengths:**

* Extensive evaluations were conducted on various deepfake detectors on various aspects.

**Weaknesses:**

* The paper isn't well written. Many times an argument is presented without being justified or citing the literature. Please see the examples provided in the "Questions" section. Table 1 and Figure 1 were not referred to by the main text of the paper.

* The motivation of the paper isn't well articulated. It is unclear from the sentence "To more comprehensively assess the detection capabilities of algorithms under various complex conditions, we have built a Deepfake Detector Assessment Platform (DAP)", why this paper is important. Please explicitly articulate the extra contributions over the three cited deepfake benchmark works in Table 1. For example, why is a new evaluation platform needed? If extra aspects like "attribute bias assessment", "adversarial attack resilience evaluation", "forgery localization accuracy evaluation" as mentioned in line 145 are important, why not find the most suitable existing platform listed in Table and extend it?

* Each subsection of Section 3 Proposed Evaluation Framework contains mostly descriptions without insights into why they were proposed as written.

* Line 215: "The platform generated a total of 5,976,145 fake images... Therefore, this evaluation can test the detection algorithm’s performance on forgery techniques that may have never been encountered before and obtain more objective generalization evaluation results." It’s unclear how merely a large number (5M) of images can achieve this goal. If a claim (which also needs to be justified) is like “a large number of deepfake algorithms spans a rich enough space that may encompass fake images generated by unseen deepfake algorithms”, then it may make more sense.

* The paper aims to offer insights into the strengths and limitations of the existing deepfake detectors, but it fails to do so by merely reporting the experimental results without analyzing why some were performing well or badly. It also didn't discuss the characteristics of deepfake detection algorithms that could have shed light on their performance. For example, line 373 reads "The highest and lowest accuracies are 80.42% and 36.52%, respectively. Only EfficientNetB4 and UCF performed relatively well compared to the other eight detectors. Additionally, the Large Multimodal Model is the second most effectively detected category by them."

* Line 358 reads: "Overall, the accuracy of the detection algorithms is generally low." It should at least compare these numbers with those in the literature to provide a sense of whether these numbers are abnormal.

* The conclusion overclaimed what the paper did, e.g., "This paper analyzes two issues" and "we have identified potential causes of these issues".

**Questions:**

Other comments:

1. Lines 67-68: "... they have several shortcomings: (1) They almost exclusively depend on public databases; (2) They lack actual tests with self-generated fake data" Line 200: "The fake data is entirely generated by the evaluation platform itself." It's unclear why exclusively depending on public data and testing with self-generated fake data are drawbacks. Please substantiate your argument.

1. Line 198: "Except for four deepfake types in Benchmark Performance Evaluation, Large Multimodal Model is included as the fifth deepfake type." Why isn’t this included in the same block as the other 4 deepfake types? Btw, there’s also a logical issue in writing, i.e., “a (large multimodal) model can’t be a “(deepfake) type”.

1. Line 215: "The platform generated a total of 5,976,145 fake images. Through the above generation pipeline, the platform simulates the complex forgery situation in the real world." Do the images include those from video frames? If so, it’s better to separately report the number of still images and the number of videos as still images and video frames have different temporal effects on human observers.

1. Figure 5's illustration and its corresponding text descriptions in Section 3.4 are unclear.

1. Line 245: "We selected the following nine types as common image disruptions: Compression, Brightness, Contrast, Flip, Rotation, Color, Sharpness, Blur, and Noise." What’s the justification or insight here? Are they from the common practice in the literature that is shown to be the most effective, or are they just some numbers (between lines 249-255) proposed by the authors? Without either of them, the paper reads like a manual and doesn’t have much scientific value or novelty.

1. Line 281: "We selected five attributes for evaluation, including Camera Angle, Gender, Ethnic Group, Expression, and Lighting Condition." Lack justification.

1. Line 409: "For better comparison, the results for the four deepfake categories are averaged in this section." It's unclear whether it allows better compression.

1. Line 412: "EfficientNetB4 and UCF remain the strongest detectors..." Insights lacking.

1. Could you comment on the paper on how easily the platform can be extended to include other deepfake detection algorithms? What efforts have been made to ensure this?

Please improve the following writing:

1. Line 38: It was mentioned at the beginning of the second paragraph of the intro section that "Unfortunately, the detection accuracy of these detectors is actually low." This reads a bit abrupt. It should either provide references to the literature or make a forecast to the experimental results of the paper to substantiate this claim.

1. Lines 43-45 are not well articulated. Why not just say they failed to generalize on unseen deepfake algorithms and distortion types? In addition, the bolded texts affect the logical flow of the paragraph.

1. Lines 52-53: "Consequently, in practical detection scenarios, besides basic accuracy, the capabilities of Forgery Algorithm Generalization, Image Distortion Robustness, and Adversarial Attack Resilience are equally important." They can be all important, but there’s no need to claim that they are “equally important.”

1. Line 146: "The only study that constructs a private dataset uses..." Please be specific about which study.

1. Line "**D**eepfake **D**etector **A**ssessment **P**latform (DAP)". Two “D”s are bolded but one “D” is in the short form. Could you figure out which “D” goes into the short form?

1. Line 173: "which covers 27 evaluation tasks..." It's unclear what these tasks are.

1. Line 191: "... the platform calculates various standardized evaluation metrics ..." Please be specific about the metrics being used.

1. The statements in lines 265, 278, 292, and 355 have logical issues. The one in line 265 reads "This section is primarily used to evaluate ..."

1. Line 318: "The platform implements 11 popular public databases, including ..." Logical flaws.

1. Line 418: "This part evaluates whether the detectors are capable of resisting adversarial attacks." "whether" should be "to what extent".

1. Figure 1: "The three bottom boxes correspond to the weaknesses of the existing DF detection algorithms. It’s logically unclear why they are not put nearer or within the “Detection” box."

---

### Official Review · Reviewer_Gfjm · 2024-11-04

**Soundness:** 3
**Presentation:** 2
**Contribution:** 2
**Rating:** 5
**Confidence:** 5

**Summary:**

This paper proposes a comprehensive Deepfake Detector Assessment Platform designed to evaluate and improve the performance of deepfake detection algorithms. The authors built a platform with 27 evaluation tasks covering six key dimensions to comprehensively evaluate deepfake detection algorithms. The platform conducts extensive experiments on public and self-built databases, considering various forgery techniques, image distortions, adversarial attacks, and attributes.

**Strengths:**

1.This paper provides a comprehensive evaluation framework, provides new directions and tools for the research of deep fake detection algorithms, and also emphasizes the importance of considering the generalization ability and resistance to adversarial attacks of algorithms in practical applications.
2.This paper conducts experiments on multiple public and self-built databases, covering a variety of forgery techniques and algorithms, and provides rich experimental data and result analysis.

**Weaknesses:**

1.This paper is insufficient in terms of theoretical analysis and method design.
2.This paper does not focus on video detection.
3.There is insufficient research on forgery detection of compressed images, which is common in social media.

**Questions:**

It is recommended that the authors provide a detailed explanation of why the algorithms work and the rationale behind them.

---

### Official Review · Reviewer_UP4U · 2024-11-08

**Soundness:** 3
**Presentation:** 3
**Contribution:** 3
**Rating:** 3
**Confidence:** 5

**Summary:**

This paper proposes a platform called Deepfake Detector Assessment Platform (DAP), to assess Deepfake detectors in terms of benchmark performance, forgery technique generalization, image distortion robustness, resilience to adversarial attacks, forgery localization accuracy and attribution bias. The paper also goes in length to discuss the approach as well as conduct extensive experiments on many datasets, which throws insights into strengths and weaknesses of current Deepfake detectors.

**Strengths:**

The paper is comprehensive and detailed on many fronts. It is very important to have a platform to assess Deepfake detectors. Currently, this is done by performance evaluations in different papers or in online media forensics challenges. However, what works on papers usually does not work very well on challenges. This paper attempts to consolidate Deepfake detectors using a platform.

The experiments, the comparisons, the datasets used, the methods compared are all very comprehensive and detailed.

**Weaknesses:**

While the problem the paper is trying to address is important, there are some key issues which need to be addressed.

Firstly, the paper builds the narrative that it is proposing a platform called Deepfake Detector Assessment Platform (DAP) for rigorous benchmarking of Deepfake Detectors. But the paper does not discuss what this platform is, what the architecture is, how this platform can be utilized by the forensics community or other parties and so on. Since the paper proposes a platform, this is very important. Though some hints are there in the Github repository such as Docker and backend-api, a section describing the system architecture of the DAP platform is definitely needed. Without this section, this paper is just a benchmark paper for Deepfake Detection.

Second, the paper gives a lot of focus to Robustness, Image Distortions and Adversarial perturbations, which are all good experiments to have. However, the paper treats distortions such as compression, smoothening, noise, blur as independent distortions. In a realistic scenario, a combination of these will usually be applied. One more experiment which randomly picks combinations of distortions and then measuring the performance metrics is needed, and this will make the experiments catering to realistic use cases.

Third, the paper focuses mainly on a large number of Deepfake datasets, which is good. But it will also be better, if the paper has a section where a large number of wild face images and/or videos are taken, and seeing how these algorithms perform on them. This will shed insights on how biased the Deepfake detection algorithms are.

**Questions:**

How do the algorithms perform on combinations of distortions and/or adversarial perturbations?

How do the algorithms perform on real world wild data (not part of datasets)?

What is the system architecture of the proposed platform?

---

### Meta-Review · Area_Chair_t22o · 2024-12-13

**Metareview:**

This paper presents a comprehensive Deepfake Detector Assessment Platform (DAP) for deepfake detection that is composed of six critical dimensions, including benchmark performance, forgery algorithm generalization, image distortion robustness, adversarial attack resilience, forgery localization accuracy, and attribute bias.
However, the reviewers raised several concerns below but the authors did not rebut:
1. System architecture of the proposed platform is not clear.
2. Performance on combination of attacks is unknown.
3. Rationale behind the proposed DAP is not clear.
4. Motivation, innovation, and writing of this paper need remarkably improvement.
Although the attempt of this work is encouraging, its current status cannot be accepted based on the high standard of ICLR.

**Additional Comments On Reviewer Discussion:**

none! The authors did not rebut at all!

---

### Decision · Program_Chairs · 2025-01-22

Reject